# Enhanced control of self-doping in halide perovskites for improved thermoelectric performance

Tianjun Liu[1,2], Xiaoming Zhao[2], Jianwei Li[3], Zilu Liu[3], Fabiola Liscio[4], Silvia Milita[4], Bob C. Schroeder [3] & Oliver Fenwick[1,2]*

Metal halide perovskites have emerged as promising photovoltaic materials, but, despite ultralow thermal conductivity, progress on developing them for thermoelectrics has been limited. Here, we report the thermoelectric properties of all-inorganic tin based perovskites with enhanced air stability. Fine tuning the thermoelectric properties of the films is achieved by self-doping through the oxidation of tin (II) to tin (IV) in a thin surface-layer that transfers charge to the bulk. This separates the doping defects from the transport region, enabling enhanced electrical conductivity. We show that this arises due to a chlorine-rich surface layer that acts simultaneously as the source of free charges and a sacrificial layer protecting the bulk from oxidation. Moreover, we achieve a figure-of-merit (ZT) of $0.14 \pm 0.01$ when chlorine-doping and degree of the oxidation are optimised in tandem.

[1] School of Engineering and Material Sciences, Queen Mary University of London, Mile End Road, London E1 4NS, UK. [2] The Organic Thermoelectrics Laboratory, Materials Research Institute, Queen Mary University of London, Mile End Road, London E1 4NS, UK. [3] Department of Chemistry, University College London, 20 Gordon Street, London WC1H 0AJ, UK. [4] Istituto per la Microelettronica e Microsistemi (IMM)-Consiglio Nazionale delle Ricerche (CNR), Via Gobetti 101, 40129 Bologna, Italy. *email: o.fenwick@qmul.ac.uk

With rapidly rising greenhouse gas emissions to the atmosphere, it is paramount to develop technologies able to generate energy at negligible cost to the environment, and to reverse the currently accelerating climatic changes. However, to successfully fulfil the transition from fossil fuels to renewable energy sources, we can no longer rely solely on existing materials, but must focus on the synthesis of other material classes with improved properties. Halide perovskites have been recognized as promising photovoltaic materials[1–3] achieving a power conversion efficiency exceeding 25%[4], due to their large absorption coefficients, high charge carrier mobilities[5] and large carrier diffusion lengths[6]. They are a highly versatile class of semiconductors, with a band gap that is tuneable through the composition of the inorganic framework, the choice of organic or inorganic cation, stoichiometry, and through self-assembly into layered structures[7–10] and nanoparticles[11]. This diversity in structure has enabled the range of applications of these materials to extend to other optoelectronic devices, including light-emitting diodes (LEDs)[12–14], X-ray detectors[15,16] and lasers[17,18].

Despite intense research on halide perovskite materials for optoelectronics, there have only been a small number of experimental studies on their thermoelectric properties, where a temperature gradient across the material can move free charge carriers and generate a thermal voltage. Thermoelectric generators can produce electrical power from temperature gradients, and to do so efficiently, must use materials possessing a high figure-of-merit, ZT:

$$ZT = \sigma\alpha^2 T/\kappa \qquad (1)$$

where $\sigma$, $\alpha$ and $\kappa$ are the electrical conductivity, Seebeck coefficient and thermal conductivity, respectively. $T$ is the temperature. Halide perovskites have an $ABX_3$ stoichiometry comprising a network of inorganic (metal-halide) octahedra with loosely bound organic or inorganic cations occupying the cavities between octahedra. These cations provide rattling modes which scatter phonons, enabling ultralow values of thermal conductivity that are now well-documented[19,20]. Combined with the high charge mobilities[5] observed in many halide perovskites, the relatively small number of experimental reports of ZT to date[21,22] in these materials is perhaps surprising.

In 2014, He et al. studied thermoelectric properties of methylammonium lead iodide (MAPbI$_3$) and methylammonium tin iodide (MASnI$_3$) by ab initio calculations[23]. They found that both materials exhibit small carrier effective mass and weak phonon-phonon and hole-phonon couplings, and predicted ZT in the range of 1–2, in-line with state-of-the-art thermoelectric materials. Shortly afterwards, Mettan et al.[21] measured the thermoelectric properties of MAPbI$_3$ and MASnI$_3$ bulk crystals[21] finding that photo-induced doping of MAPbI$_3$ and chemical doping of MASnI$_3$ improved ZT. They concluded that MAPbI$_3$ would be a good candidate for the thermoelectric applications due the high hole mobility, large Seebeck coefficient and a remarkably low thermal conductivity. However, the low charge carrier density is a barrier to further development. Low charge carrier density in these materials is a product of ionic compensation of charged point defects[24], as well as a defect tolerant electronic structure arising from bonding orbitals at the conduction band minimum, and antibonding orbitals at the valence band maximum[25]. The resulting low density of deep defects is an excellent feature for optoelectronic applications since defects can quench electroluminescence in LEDs or lead to recombination of photo-generated charges in solar cells. On the other hand, thermoelectric applications require charge densities typical of heavily doped semiconductors $\sim10^{18}$–$10^{20}$ cm$^{-3}$, and doping would usually come from defect sites, such as substitution of a higher valency metal atom on the perovskite B-site[26]. This makes development of halide perovskites for thermoelectrics challenging.

An exception are the lead-free tin halide perovskites, such as the cubic perovskite CH$_3$NH$_3$SnI$_3$, which shows metallic conductivity[27]. Takahashi et al.[28] noted that high conductivity in CH$_3$NH$_3$SnI$_3$ bulk crystals arises from a self-doping process through the oxidation of Sn$^{2+}$ to Sn$^{4+}$[28]. In 2017, Lee et al. reported the ultralow thermal conductivity of a single CsSnI$_3$ nanowire and a ZT of 0.11 at 320 K[22], whilst Saini et al. report a ZT in thin films of 0.137 at 292 K[29]. However, the underlying physical mechanisms that determine thermoelectric performance of halide perovskite materials are not completely understood, and significant issues remain unaddressed such as identification of ZT optimisation strategies.

In this work, we develop a series of vacuum thermal evaporation methods to fabricate lead-free CsSnI$_3$ perovskite thin films. We find air stability and electrical conductivity of our films to be highly tuneable by the deposition process with films formed by sequential deposition of the precursors yielding electrical conductivity 25 times that of films formed by co-evaporation of the same precursors. Compared with organic-inorganic hybrid perovskites, all-inorganic halide perovskites present significant improvements in thermal stability[30,31], but we enhance this further by developing a method to substitutionally dope chlorine into the perovskite structure in the top 10 nm of our films. A by-product of air exposure is the oxidation of Sn$^{2+}$ to Sn$^{4+}$ (self-doping), and we exploit this in a controlled manner to fine tune the electrical conductivity and thermoelectric properties of the mixed halide CsSnI$_{3-x}$Cl$_x$ perovskite thin films. We quantify the Sn oxidation states as a function of depth in mixed halide perovskite films using Auger electron spectroscopy, showing an unusual mechanism whereby an oxidised top surface-layer (<10-nm thick) is responsible for electrical doping the underlying film (250–300-nm thick). In this surface doping configuration, the dopants do not disrupt the crystal structure in the part of the film responsible for charge transport. In fact our Seebeck measurements indicate that the electrical doping levels in our films rise in tandem with the amount of Cl substituted in the top layers, showing that chlorine doping is simultaneously providing free charges to the system and acting as a sacrificial surface layer that slows oxidation of the bulk. We furthermore verify the applicability of the Wiedemann-Franz law in this class of materials with a value of the Lorenz number close to the Sommerfeld value, and achieve a ZT of 0.14 at 345 K upon simultaneous optimisation of the degree of Cl-doping and the degree of oxidation.

## Results

**Thermal vapour deposition of CsSnI$_3$ perovskite films.** Past approaches to synthesize CsSnI$_3$ have included solution processing by spin-coating[31] and growth of single crystals[22,32]. In our case we have developed thermal vapour deposition approaches in order to achieve a high quality of films with fine control over morphology and composition.

Starting from the precursors caesium iodide (CsI) and stannous iodide (SnI$_2$), we developed three different vacuum deposition methods to prepare the perovskite films: co-evaporation, sequential deposition and seed layer plus sequential deposition (SLS) (Fig. 1a–c). For the co-evaporation process (Fig. 1a), the perovskite was obtained directly from simultaneous vacuum thermal evaporation of the two precursor materials (SnI$_2$ and CsI). For the sequential deposition method (Fig. 1b), CsI and SnI$_2$ were sequentially deposited to form a bilayer film which was then baked to form the perovskite. For the SLS method (Fig. 1c), a co-evaporated perovskite seed layer was introduced before sequential deposition, and the film was post-baked to form the perovskite

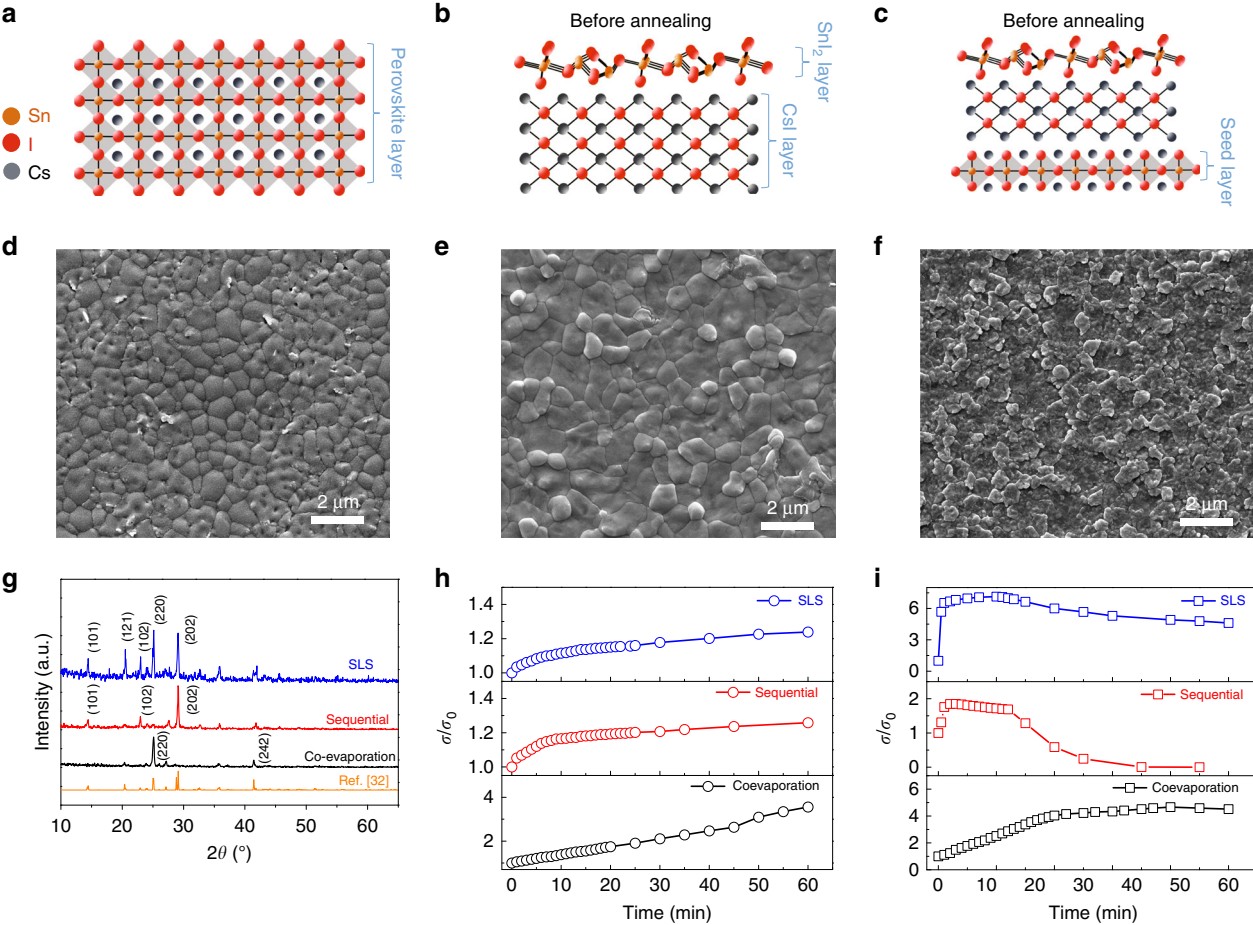

**Fig. 1 Morphology, crystal structure and electrical conductivity of CsSnI₃ films. a–c** Schematics of film deposition of the co-evaporation, sequential and SLS methods, respectively (before any annealing steps). **d–f** Corresponding scanning electron microscopy (SEM) images of the films after any annealing steps. **g** X-ray diffraction spectra of three types of thin film with lattice plane indices of the most prominent peaks in each case. Electrical conductivity of three types of perovskite film in nitrogen atmosphere (**h**) and in air (**i**).

structure. Co-evaporated films were mirror-black, characteristic of the CsSnI₃ perovskite, whilst sequentially deposited and SLS films were red-brown, but became mirror-black after baking at 170 °C in nitrogen atmosphere (Supplementary Fig. 1). Scanning electron microscopy (SEM) revealed the dense polycrystalline morphology of the vapour-deposited CsSnI₃ thin films (Fig. 1d–f). Sequential deposition produced perovskite thin films with around 1 μm diameter grains, which were larger than grains in the co-evaporated perovskite thin film (300–500 nm diameter). The SLS perovskite thin films also contained sub-micron grains, yet with a rougher surface morphology. As shown in Fig. 1g, X-ray diffraction patterns of CsSnI₃ films made by all three deposition procedures showed features of the orthorhombic black phase, B-γ[32] of CsSnI₃, with peaks at 25.02° and 29.15° ($2\theta$) corresponding to (220) and (202) planes, respectively. The sequentially processed films have a dominant peak at 29.15°, showing a preferred orientation of the (202) crystal plane parallel to the substrate. On the other hand, the co-evaporated films present preferential orientation of the (220) plane parallel to the surface. In the case of the SLS processed films, multiple peaks were observed, including both (220) and (202), indicating mixed orientations of crystallites in the film. The films are present in the B-γ phase regardless of deposition method, which is confirmed with grazing-incidence X-ray diffraction (GIXRD) experiments (Supplementary Fig. 2), and there was no evidence of diffraction peaks associated with Cs₂SnI₆ or the precursor materials. The thickness of all films studied was between 250 and 300 nm.

**Electrical conductivity and stability**. To characterise the electrical stability of our films, we performed time-dependent electrical conductivity measurements both in inert atmosphere (N₂ glovebox) and in air. CsSnI₃ thin films from all three deposition methods showed high stability when tested in a N₂ atmosphere, in fact showing a modest increase in electrical conductivity over a period of 1 h (Fig. 1h). In total over that period a reproducible increase in conductivity by a factor of 3.6 was observed for co-evaporated films, 1.3 for sequentially evaporated and 1.2 for SLS films, reaching maximum conductivities of $8.5 \times 10^{-3}$, 7.3 and 6.8 S cm⁻¹, respectively. When the thin films were exposed to air, $\sigma$ increased by a factor of 2 to 7 in all cases (Fig. 1i), which would be expected from a self-doping process during oxidation of Sn²⁺ to Sn⁴⁺[33,34]. $\sigma$ of co-evaporated thin films continuously increased for 45 min, while $\sigma$ of sequentially deposited films increased for just 5 min before degradation caused a rapid decrease. The thin films deposited by the SLS method can sustain increases in $\sigma$ for 11 min, reaching a value 7 times the initial one and remain reasonably stable afterwards, showing only 30% reduction in $\sigma_{max}$ over the following 50 min. CsSnI₃ thin films deposited by SLS show a similar maximum electrical conductivity (37.1 S cm⁻¹) to sequentially deposited films (32.2 S cm⁻¹), which is ~25 times higher than the maximum for co-evaporated films (1.2 S cm⁻¹), a value we consider too low for thermoelectric applications. Co-evaporated films with dominate orientation (220) therefore show the best air stability but lowest electrical conductivity. Sequentially deposited films with dominate orientation (202) show a

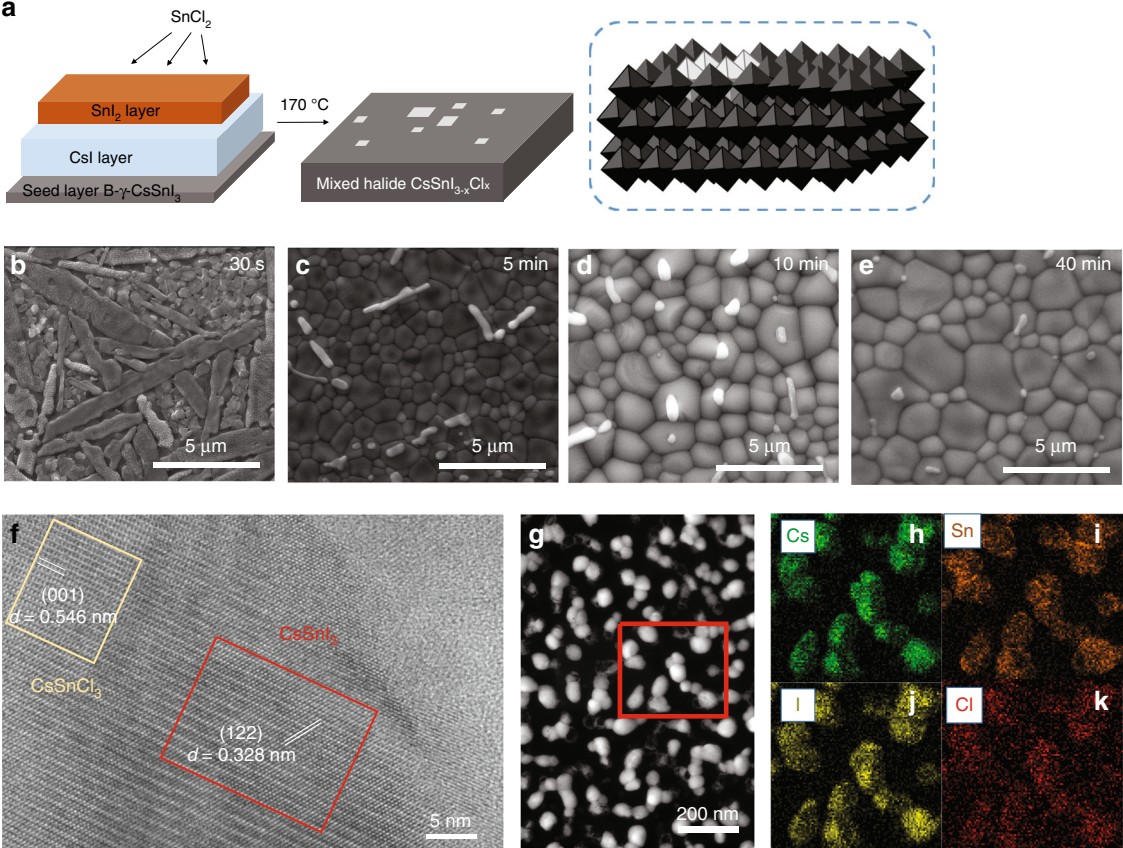

**Fig. 2 Mixed halide CsSnI$_{3-x}$Cl$_x$ perovskite morphology, structure and elemental distribution. a** Schematic of the mixed halide perovskite deposition method. The white squares represent the CsSnCl$_3$ structures in the top layers of the B-γ-CsSnI$_3$ perovskite films. **b–e** SEM images of the morphological development of mixed halide perovskite films as a function of baking time. **f** High-resolution transmission electron microscopy (HRTEM) of mixed halide perovskite structures. The yellow square and red square correspond to CsSnCl$_3$ and CsSnI$_3$ crystal lattices, respectively. **g** STEM-HAADF image of isolated grains of mixed halide CsSnI$_{3-x}$Cl$_x$ formed on an amorphous carbon support and (**h–k**) STEM-EDS elemental mapping in the area denoted by the red square in (**g**).

poor air stability despite their larger grain sizes, but do have higher electrical conductivity. Previous work has shown that grain orientation can have a significant effect on degradation rates of halide perovskite films[35] and this is likely to be the case here. Since SLS films have enhanced stability compared with the sequentially deposited films, we chose SLS produced films as the platform from which to optimise the thermoelectric properties of CsSnI$_3$ perovskites.

To further improve the air stability of SLS perovskite thin films, chloride was introduced in the deposition process, as mixed halide perovskites are known to exhibit improved air stability over analogous single-halide materials[31,36]. This was done by thermal deposition of a thin layer (<20 nm) of tin chloride (SnCl$_2$) on top of the 250–300-nm thick SLS films prior to thermal annealing (schematic in Fig. 2a). Deposition was followed by baking under nitrogen atmosphere at 170 °C. An initial SEM investigation revealed an elongated grain structure on the top surface of our films (Fig. 2b) which was attribute to a pure SnCl$_2$ phase. As baking progresses (Fig. 2c), the typical polycrystalline perovskite morphology with polygonal grains emerges, although a small number of the elongated crystals remain on top. On further baking, the remaining elongated crystals show reduced aspect ratio and the underlying perovskite grains merge into micron-sized features (Fig. 2d), until after 40 min of baking (Fig. 2e), there was little evidence of the elongated crystals at all. The mixed halide films have XRD features similar to CsSnI$_3$ with an absence of peaks that could be assigned to SnCl$_2$ or CsSnCl$_3$

(Supplementary Figs. 3 and 4). It should be noted that SnCl$_2$ can sublime at 170 °C, so we used SEM and STEM combined with energy-dispersive X-ray spectroscopy (EDS) (Supplementary Fig. 5 and Fig. 2f–k, respectively) to confirm residual Cl incorporation into our samples. Furthermore, high-resolution transmission electron microscopy (HRTEM) of a single grain of our mixed halide perovskite (Fig. 2f), showed two regions of different crystal lattices (marked with red and yellow squares). The lattice spacing of 0.328 nm measured in the red region corresponds to the CsSnI$_3$ crystal (122) plane, whilst the lattice spacing of 0.546 nm measured in the yellow region corresponds to the (001) plane of the CsSnCl$_3$ cubic lattice. This indicates a degree of nanoscale phase separation between chlorine-rich and iodine rich phases within perovskite grains. The absence of CsSnCl$_3$ features in the XRD spectra is due to the low concentration of Cl in our films. Corroborating evidence for the incorporation of chlorine into perovskite structures is provided by X-ray photoelectron spectroscopy (XPS), with the Cl 2$p$ peak of our mixed halide films showing a significant broadening compared with SnCl$_2$ (Supplementary Fig. 6)[31]. Moreover, we used XPS to get a depth profile of the Cl concentration in our films (Supplementary Fig. 7), finding that Cl was present in the top layer, penetrating only a few nanometres into the bulk. We could not detect any chlorine by XPS at depths larger than 10 nm from the film surface. In what follows, we studied 0.5, 1, 3 and 5% SnCl$_2$ mixed halide CsSnI$_{3-x}$Cl$_x$ perovskite films. The percentage we use refers to the mass of SnCl$_2$ relative to SnI$_2$ in our thin films

before the baking step. The final atomic % of Cl in the film will be much lower.

To demonstrate the improved stability of our mixed halide perovskite films, we studied the quenching of the optical absorption peak at 420 nm (Supplementary Fig. 8). 5% Cl-doped SLS films show enhanced stability, with just 3% quenching of the 420 nm peak after 100 min air exposure, whereas SLS CsSnI$_3$ films without Cl-doping showed a 40% quenching of the peak under the same conditions. In fact, in terms of their optical properties, the 5% Cl-doped SLS films are more stable than undoped co-evaporated films, showing less than half of the quenching of the absorption after 500 min in air.

**Quantitative analysis of Sn oxidation states.** As the origin of high conductivity in tin halide perovskites comes from hole doping due to the oxidation of Sn$^{2+}$ to Sn$^{4+}$, we used XPS analysis to probe the oxidation state of Sn in our films. Shifts in the Sn $3d_{5/2}$ peak are relatively modest as a function of oxidation state (Supplementary Fig. 9), so we focussed on the Auger region of the spectrum. Since Auger electron spectroscopy (AES) probes three-electron process, it is a much more sensitive measure of oxidation state. We did this as a function of depth in CsSnI$_{3-x}$Cl$_x$ films (1% SnCl$_2$) which had undergone a short air exposure (Fig. 3a). The Sn MNN AES spectrum shows a broad line shape, including several Sn MNN peaks (fitting curves labelled $a$, $b$, $c$ and $d$, with details in Supplementary Table 1). In the Sn$^0$ metal M$_5$N$_{4,5}$N$_{4,5}$ AES spectrum reported by Barlow et al.[37], $^1$S$_0$ has a peak at a kinetic energy of 421.2 eV, and showed a large broadening after oxidation. In our case, the $^1$S$_0$ peak (fitted curve $a$) is broad, confirming the absence of Sn$^0$ states. Fitted curve $b$ (425–430 eV) includes multiplet splitting of the $^1$G$_4$, $^3$P$_2$, $^3$F$_{2,3}$ and $^3$F$_4$ states (Supplementary Table 1). This broad peak shifts to higher kinetic energy with increasing etching depth, corresponding to oxidation in the top layer compared with the bulk[38–40]. This is even more evident from peak $c$, which is linked to Sn$^{4+}$ states[38–40] and is prominent at the surface, but disappears completely within an etching depth of 7.5 nm (Fig. 3b).

The modified Auger parameter ($\alpha'$) can be used for a more robust identification of chemical states of elements in molecules or solids, and is not susceptible to shifts caused by sample charging[38–41]. It is defined as the sum of the kinetic energy of a core-core-core Auger line, $E_k$, and the binding energy, $E_b$, of a core electron, $\alpha' = E_k + E_b$ and can be viewed more intuitively if plotted in a Wagner format[42] (a plot of $E_k$ versus $E_b$ recorded from all chemical states of the atom). The Wagner plot in Fig. 3c, combines our own data (as a function of depth) with some literature references for Sn$^0$, Sn$^{2+}$ and Sn$^{4+}$ states[39,40,43], and clearly illustrates a mixture of Sn$^{2+}$ and Sn$^{4+}$ oxidation states in our film (Sn core binding energy, Auger kinetic energy and Auger parameters detailed in Supplementary Tables 2–3 and Supplementary Fig. 10)[44]. Moreover, there is a progressive change from majority Sn$^{4+}$ states at the surface of the film to Sn$^{2+}$ at a depth of 10 nm, further evidence that the oxidation process only occurs in the top 7.5 nm of the film, the same region of the film that incorporates chlorine dopants.

From this, we can conclude that the top surface layer of the mixed CsSnI$_{3-x}$Cl$_x$ acts as a sacrificial layer where initial oxidation occurs. This layer provides hole doping to the bulk (vide infra) from the surface Sn$^{4+}$ species. This mechanism, which separates the doping layer from the transport region, minimises the structural impact of doping on charge mobility, and enables our mixed halide perovskite structure to present high electrical conductivity whilst retaining a reasonable degree of air stability.

**Thermoelectric properties of CsSnI$_{3-x}$Cl$_x$ thin films.** We performed thermoelectric property measurements as shown in Fig. 4a–f. The temperature dependence of $\sigma$ and the sign of $\alpha$ for 1% Cl-doped CsSnI$_{3-x}$Cl$_x$ in the range 290–360 K (Fig. 4a, b) indicates band-like transport and that the majority charge carriers are holes, as reported previously for CsSnI$_3$ single crystal nanowires[22], validating the high quality of our mixed halide perovskite films. $\alpha$ increases approximately linearly with temperature due to the shift of the Fermi level away from the valence band, following the Fermi-Dirac distribution within the mobility edge model of the Seebeck coefficient for a heavily doped semiconductors[45].

We can fine tune the electrical conductivity of our films by exposing them to air (3 min at a time) to further oxidise Sn$^{2+}$ to Sn$^{4+}$. At room temperature, initial electrical conductivity, $\sigma_0$, was

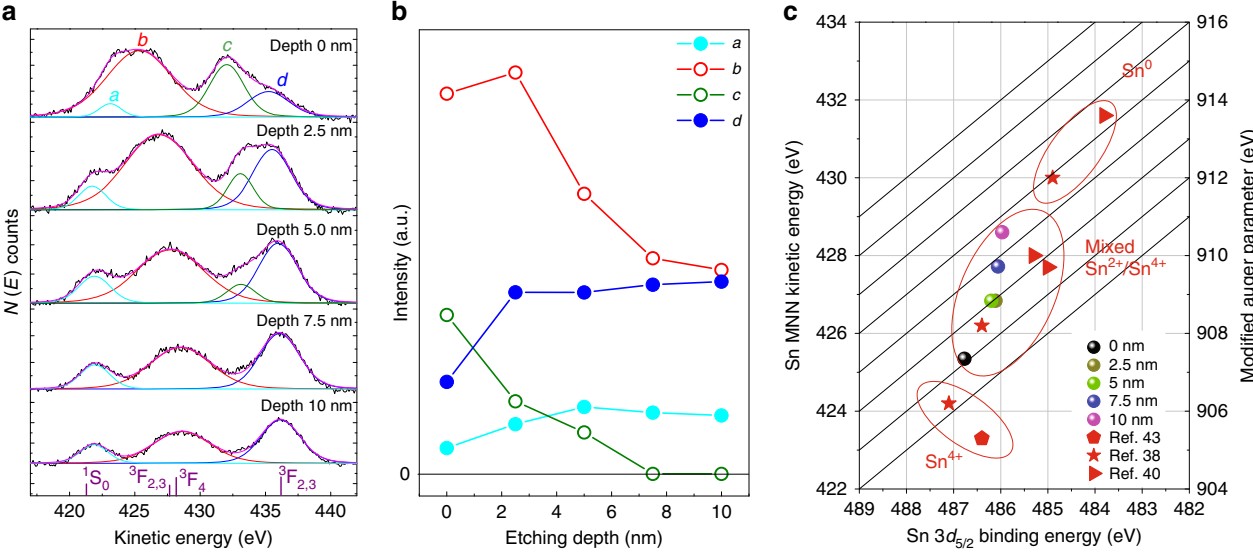

**Fig. 3 Sn oxidation state in 1% Cl-doped CsSnI$_{3-x}$Cl$_x$ perovskite thin films. a** Auger electron spectra of Sn MNN at different etching depths from 0 to 10 nm. **b** Photoelectron counts of fitted curves in (**a**) as a function of etching depth. **c** Sn $3d_{5/2}$ Wagner plot with the modified Auger parameter of our samples (circles) and reference values for Sn$^0$, Sn$^{2+}$ and Sn$^{4+}$.

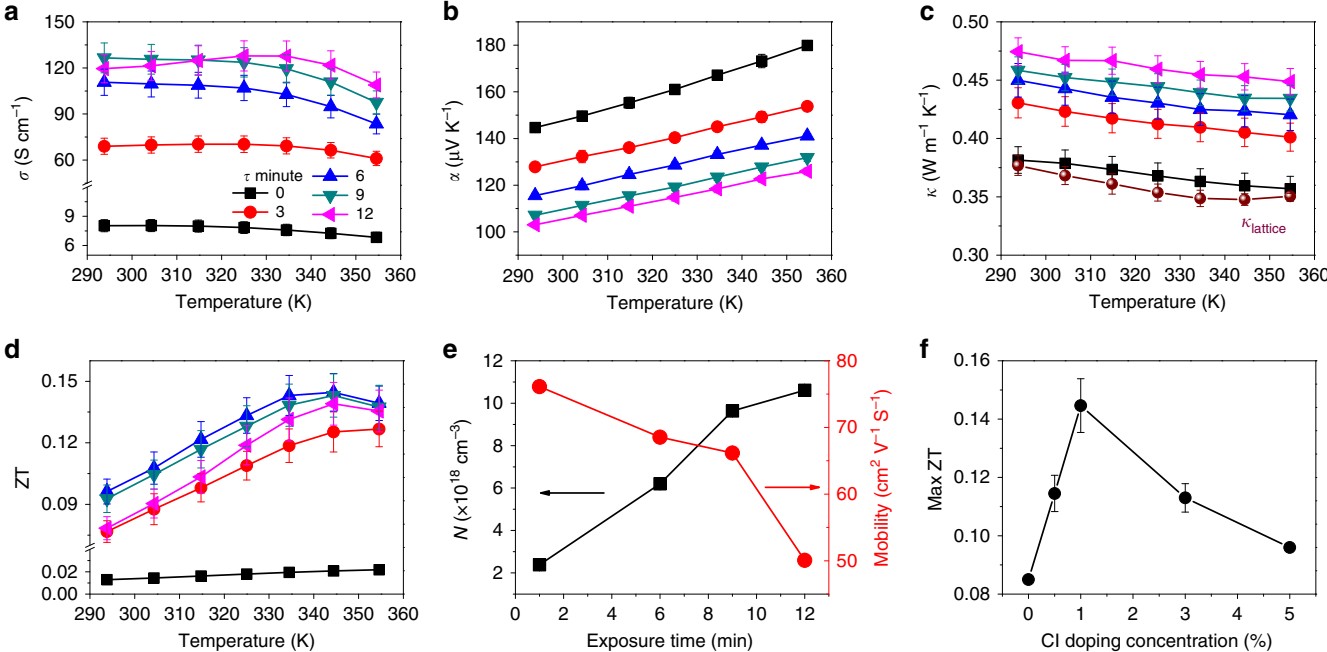

**Fig. 4 Thermoelectric properties of 1% mixed CsSnI$_{3-x}$Cl$_x$ perovskite thin films.** Temperature dependence of electrical conductivity $\sigma$ (**a**), Seebeck coefficient $\alpha$ (**b**), thermal conductivity $\kappa_{total}$ (**c**) and figure-of-merit, ZT (**d**). The differently coloured curves represent different degrees oxidation according to the legend in plot (**a**). Error bars in (**b**) are too small to be visible, typically representing <2% of the value. **e** Charge carrier density and Hall mobility as a function of air exposure time. **f** Maximum figure-of-merit, ZT, as a function of SnCl$_2$ incorporation.

$8.0 \pm 0.6$ S cm$^{-1}$ (Fig. 4a) and it dramatically increased to $69.0 \pm 5.2$ S cm$^{-1}$ after a further air exposure ($\sigma_3$), and saturated at $126.5 \pm 9.7$ S cm$^{-1}$ after 9 min air exposure ($\sigma_9$). Further air exposure lead to a slower, but steady decrease in electrical conductivity ($\sigma_{12} = 119.6 \pm 9.2$ S cm$^{-1}$). The dependence of $\sigma$ on air exposure time comes from the competition between the enhanced charge carrier concentration owing to self-doping from Sn$^{4+}$ species and reduced carrier mobility owing to defects caused by air exposure, which can take the form of degradation in the bulk or at the grain boundaries during the oxidation process, or even increased ionised impurity scattering.

The Seebeck coefficient, $\alpha$, in Fig. 4b shows a steady decrease with exposure time (at room temperature, $\alpha_0 = 144.7 \pm 1.5$ μV K$^{-1}$, and after 12 min air exposure $\alpha_{12} = 103.0 \pm 1.0$ μV K$^{-1}$), consistent with a steady increase in the charge concentration shifting the Fermi energy level, $E_f$, towards the valence band. The observation that the Seebeck coefficient continues to decrease with air exposure when the electrical conductivity has already peaked is further evidence that the degradation in conductivity after extended exposure to air is due to mobility lowering processes. To verify this hypothesis, we used Hall measurements to determine the charge carrier concentration as a function of air exposure, showing an increase with air exposure from $2.38 \times 10^{18}$ to $1.06 \times 10^{19}$ cm$^{-3}$ after 12 min (Fig. 4e). Meanwhile, the Hall mobility decreases from an initial value of 76.1 to 50.1 cm$^2$ V$^{-1}$ s$^{-1}$ after oxidation. We note that in Lee et al.'s work[22], lower $\alpha$ (79 μV K$^{-1}$) at room temperature with higher $\sigma$ (282 S cm$^{-1}$) indicates a higher level of self-doping, whereas our control of oxidation level allows us to precisely tune the $\alpha/\sigma$ ratio and ultimately optimise ZT.

The measured temperature-dependent thermal conductivity for 1% SnCl$_2$ perovskite thin films is presented in Fig. 4c. At room temperature after a minimal air exposure of 30 s, the thermal conductivity is $0.38 \pm 0.01$ W m$^{-1}$ K$^{-1}$ and it increases with air exposure to $0.47 \pm 0.01$ W m$^{-1}$ K$^{-1}$. To obtain the lattice thermal conductivity $\kappa_{lattice}$ from the measured $\kappa_{total}$ ($= \kappa_{lattice} + \kappa_{electronic}$), we plotted $\kappa_{total}$ as a function of $\sigma$ (Supplementary

Fig. 11). The electronic thermal conductivity, $\kappa_{electronic}$, is described by Wiedemann–Franz law ($\kappa_{electronic} = \sigma L T$), enabling us to determine $\kappa_{lattice}$ and the Lorentz number, $L$, from the intercept and slope, respectively, of a linear fit. We note that since electrical doping is provided by a thin (<10 nm) surface layer, we can assume that the lattice thermal conductivity in the bulk of the film is not strongly affected by the doping process, which is a requirement for this analysis. We found $\kappa_{lattice} = 0.38 \pm 0.01$ W m$^{-1}$ K$^{-1}$ at room temperature, which is consistent with Lee's work ($0.38 \pm 0.04$ W m$^{-1}$ K$^{-1}$), and extract a Lorentz number of $(2.40 \pm 0.33) \times 10^{-8}$ W Ω K$^{-2}$ at room temperature or an average of $(2.26 \pm 0.13) \times 10^{-8}$ W Ω K$^{-2}$ over the full temperature range, close to the Sommerfeld value for free electrons. Furthermore, we can confirm that polycrystalline CsSnI$_{3-x}$Cl$_x$ thin films exhibit a temperature dependence of $\kappa_{lattice}$ that is consistent with the Calloway model[46], with Umklapp scattering processes dominating in this temperature range, as has been reported for methylammonium lead iodide perovskites[21,47].

Finally, the thermoelectric figure-of-merit, ZT, of our CsSnI$_{3-x}$Cl$_x$ perovskite films increases for oxidation time, $\tau$, in the range 0–6 min, and then decreases for $\tau$ more than 6 min, as shown in Fig. 4d. The largest ZT is $0.14 \pm 0.01$ at 345 K for 1% CsSnI$_{3-x}$Cl$_x$, a factor of 7 higher compared with that of the $\tau = 0$ sample (ZT = 0.02 at 355 K). The figure-of-merit shows a 32% reduction after 10 h in air, and a 30% reduction after 10 days storage in nitrogen atmosphere (Supplementary Figs. 12 and 13). We also note that thinner films showed higher electrical conductivities, but no improvement in ZT (Supplementary Fig. 14). This high degree of control over ZT through tuning of $\sigma$ and $\alpha$ indicates the effectiveness of self-doping in the thermoelectric performance of Sn-halide perovskites. Interestingly, the maximum ZT is a function of the degree of Cl-doping (Fig. 4f), with a sharp increase of ZT$_{max}$ from 0.07 at 0% Cl to a peak of ZT$_{max} = 0.14 \pm 0.01$ at 1% Cl and steady decrease upon further Cl-inclusion. In parallel, we observe that

the Seebeck coefficient decreases as a function of Cl-doping (Supplementary Figs. 15, 18), implying that the more heavily Cl-doped the films are, the higher the charge carrier density that is achieved. This is further evidence that the chlorine-rich surface layer is acting not only as a protective layer slowing down oxidation of the underlying CsSnI$_3$, but also as a sacrificial source of holes in this system that are donated from the surface to the bulk. This separation of the dopants from the charge transport channel prevents the introduction of scattering defects in the transport channel which can reduce charge mobility, and is the reason that our Cl-doped films can achieve up to four times the maximum electrical conductivity of our pristine CsSnI$_3$ films (Fig. 1 and Supplementary Fig. 18).

## Discussion

Our work sheds light on optimisation strategies of halide perovskites for thermoelectrics, with wider implications for the development of halide perovskite films with targeted properties across other application areas such as photovoltaics, photodetectors, thin film transistors and light-emitting diodes. We have demonstrated a number of thermal vapour deposition methods for the formation of high quality CsSnI$_3$ thin films from its precursor materials. These films are self-doping through oxidation of Sn$^{2+}$ to Sn$^{4+}$, and we have shown that the stability and electrical conductivity of the films is highly dependent on whether a sequential or co-evaporation process is adopted, with the former offering higher electrical conductivity and the latter higher stability. For this reason, we developed a hybrid of the two approaches (SLS) to offer a suitable platform from which to optimise thermoelectric properties. Beyond this and building on knowledge that mixed halide approaches can improve atmospheric stability of halide perovskites, we adopted a unique approach to chlorine doping of our CsSnI$_3$ films, that results in substitution of chlorine into a perovskite crystal lattice in a region <10 nm from the surface. We have shown that the Cl-dopants not only enhance stability but simultaneously act as a sacrificial source of free charges. The electrical doping is therefore coming from the outer atomic layers of the film, but dopes to the pristine bulk, thus dividing the film into a thin doping layer and a thicker charge transport channel and ensuring that the introduction of dopants does not degrade mobility in the transport channel.

The accessible free charge carrier concentration is therefore determined by the Cl concentration and by tuning this in combination with the degree of oxidation, we have optimised thermoelectric performance, achieving $ZT = 0.14 \pm 0.01$, and verified that the Wiedemann-Franz law is valid in these materials with a Lorenz number similar to the Sommerfeld value for free electrons. These results are important in identifying routes to develop the halide perovskite class of materials for thermoelectric applications, but the process of their optimisation for thermoelectrics has revealed a deeper understanding of their thermal and electrical transport properties, as well as strategies for controlled doping, which has implications across all areas of their application. The potential advantages of halide perovskite materials for thermoelectrics are elemental abundance as well as mechanical flexibility, solution processability and large area scalability. Finally, we note that the stability of this tin based perovskite material could be further improved by adding a strong reductant with favourable Goldschmidt tolerance into the structure, pursuing layered structures, surface passivation or adopting mixed metal approaches.

## Methods

**Film deposition**. We present three type of evaporated films: co-evaporated, sequentially evaporated and seed layer plus sequential deposition (SLS). For co-evaporated films, tin (II) iodide (SnI$_2$, 99.99%, Sigma-Aldrich) and caesium iodide (CsI, 99.99%, Sigma-Aldrich) were simultaneously deposited at 10$^{-7}$ mbar. The deposition rate was 1 Å s$^{-1}$ for SnI$_2$ (achieved with a crucible temperature of 160 °C) and 3 Å s$^{-1}$ for CsI (achieved with a crucible temperature of 430 °C). The mirror-black films were directly obtained from the co-evaporation methods without annealing. For the sequential deposition methods, SnI$_2$ was thermally evaporated at 10$^{-7}$ mbar at 2 Å s$^{-1}$ (170 °C), followed by CsI at 6 Å s$^{-1}$ (450 °C). The initially red-brown thin films were removed from the vacuum chamber for baking at 170 °C in nitrogen atmosphere. Upon baking, the appearance of the films became mirror-black indicating that CsSnI$_3$ thin films were successfully fabricated. For SLS films, a 50 nm co-evaporated layer was first deposited as a seed layer. Above the seed layer, followed a layer deposited by the sequential method without breaking vacuum. Dark brown films were obtained from the SLS method, forming mirror-black CsSnI$_3$ films upon baking at 170 °C. For mixed halide perovskite samples, tin (II) chloride (SnCl$_2$, 99.99%, Sigma-Aldrich) was evaporated at 0.5 Å s$^{-1}$ (achieved with a crucible temperature of 130 °C) on top of SLS films (which had not been baked) without breaking vacuum. The mixed halide films were also baked at 170 °C in nitrogen atmosphere to form mirror-black mixed halide perovskite films.

**Scanning electron microscopy**. The surface morphology of the films was performed on a field-emission scanning electron microscope (FEI Inspect-F).

**Optical absorption**. UV-Vis absorption spectra were measured with Shimadzu UV-2600 spectrophotometer, using 10 min intervals for time-dependent air stability studies.

**X-ray diffraction**. X-ray diffraction was performed on a Siemens D5000 X-Ray Powder diffractometer using a Cu Kα source ($\lambda = 1.54$ Å).

**Grazing-incidence X-ray diffraction**. GIXRD measurements were performed at the XRD1 beamline of the ELETTRA synchrotron facility in Trieste (Italy). The X-ray beam had a wavelength of 0.7 Å and a beam size of 200 × 200 μm$^2$. 2D-GIWAXS images were collected by using 2 M Pilatus silicon pixel X-ray detector (DECTRIS Ltd.) positioned perpendicular to the incident beam, at a distance of 260 mm from the sample. The grazing incident angle was fixed at $\alpha_i = 0.5°$ to probe the full thickness of the film.

**Scanning transmission electron microscopy-energy-dispersive X-ray spectroscopy**. Transmission electron microscopy (TEM) and high-resolution TEM imaging was carried out on a Tecnai G$^2$ F20 S-TWIN at 200 kV. High angle annular dark field scanning transmission electron microscopy (HAADF-STEM) imaging and energy-dispersive X-ray spectroscopy (EDS) elemental mapping were performed on a JEM-ARM 200F at 200 kV. The TEM specimen fabrication was by the evaporation process of CsSnI$_{3-x}$Cl$_x$ perovskite mentioned in the Film deposition section onto a copper grid with an amorphous carbon film on top.

**X-ray photoelectron spectroscopy**. XPS was performed on Thermo Scientific K-Alpha X-ray photoelectron spectrometer with a monochromatic Al Kα X-ray source under high vacuum ($2 \times 10^{-8}$ mbar). Etching of the films for depth profiling was by in situ sputtering at room temperature using a beam of 3 keV Ar$^+$ ions. The etching depth profile was calculated from the etching time required to etch through to the silicon substrate. Fitting was performed on the CasaXPS package, incorporating Voigt line shapes and a Shirley background.

**Thermoelectric properties measurement**. In-plane thermoelectric properties ($\sigma$, $\kappa$ and $\alpha$) were measured simultaneously on the same sample with a Linseis Thin Film Analyser (described elsewhere[48–50]). In this measurement geometry, in-plane thermal conductivity is measured on a suspended SiN membrane by a 3-ω method. Electrical conductivity is measured by the van der Pauw method with four needle like electrodes at the four corners of the films. The Seebeck coefficient measurement uses a thermometer and a heater on the suspended membrane to achieve a temperature gradient (schematised in the Supplementary Fig. 19). Samples fabricated in the glovebox were transferred to the Linseis Thin Film Analyser with <2 min exposure to air. The humidity in lab was around 40%. All measurements were performed under vacuum and in the dark. Hall effect measurements were performed on PPMS-9 from Quantum Design Inc. When we wished to partially oxidise the films, the measurement chamber was refilled with air to atmospheric pressure for a designated time before pumping down again for the next measurement.

## Data availability

The data that support the findings of this work are available from the corresponding author on request.

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

## Acknowledgements

The research was financed under O.F.'s Royal Society University Research Fellowship (UF140372). B.S. acknowledges financial support by the British Council (Grant No: 337323). T.L., X.Z., J.L. and Z.L. were supported by the Chinese Scholarship Council (CSC).

## Author contributions

T.L. performed the experimental work on film deposition, structure and thermoelectric property characterization. X.Z. and J.L. performed the XRD measurements and data analysis. T.L., Z.L. and B.S. performed the XPS and AES measurements. F.L. and S.M. performed GIWAX measurements. This project was conceived and planned by O.F. and T.L and supervised by O.F. The paper was written with contributions from all authors.

## Competing interests

The authors declare no competing interests.
