## [Peer Review File · Nature Communications]

Reviewers' comments:

Reviewer #1 (Remarks to the Author):

In the present work, Fenwick et al. explore all-inorganic perovskite CsPbI₃ for thermoelectric application by self-doping and addition of chlorine to further enhance the stability and performance. A ZT of 0.14 was recorded which is claimed to be the highest for halide-perovskite thermoelectrics. While this is one of the most detailed work on halide perovskite thermoelectrics, there is still a long way before appreciable ZT and stability can be achieved for halide perovskite thermoelectrics. Below are some detailed comments for a major revision.

1. Thermoelectric performance of CsSnI₃ in the form of nanowire as well as the self-doping process in Sn-based perovskites are already reported as mentioned in the manuscript. Another major reservation about the present work is a recent report (<https://doi.org/10.1557/adv.2019.279>) on CsSnI₃ thin films where a ZT of 0.137 was achieved which is equal to the present work after error estimations without any sample modification. Authors need to compare their work with prior reports and highlight the unique aspects in a separate paragraph.
2. Why different degree of oxidation and degradation was observed for the films deposited by different techniques even though the composition was same? Does orientation plays an important role in conductivity enhancement and degradation in air?
3. On page 6, authors mention that films deposited by SLS method shows the best stability and that is why they chose this sample for further investigation. But if we look at Figure 1-i, it appears that conductivity of co-evaporated film is more stable after 20 minutes of air exposure. In addition, the morphology of co-evaporated films are significantly better than other films.
4. Authors claim that the surface doping technique and inclusion of Cl enhances the stability of CsSnI₃. So it is imperative that performance stability of the best sample measured under nitrogen and air over the course of few hours/days be included.
5. Authors need to report errors in their measurements for Seebeck and thermal conductivity. Since authors mentioned that the thickness of the films were between 250-300 nm, this range will also contribute to the error estimations of different parameters. This will induce an error in the final value of ZT which should be reported.
6. What is the carrier density before and after doping since it is an important parameter in the present work.
7. Authors need to give more details of how they extracted the Lorentz number. Further, for Figure S11, authors have plotted multiple data points of thermal conductivity vs electrical conductivity for a single temperature. How different values of electrical and thermal conductivity were achieved at a single temperature? Is this achieved by different degree of oxidation of the sample? More details should be mentioned.

Reviewer #2 (Remarks to the Author):

In this manuscript, Liu T. et al. reported a study on the thermoelectric properties of vacuum deposited CsSnI₃-xCl_x perovskites via different procedures. With the surface oxidation of Sn²⁺ to Sn⁴⁺, the electrical conductivity of the bulk film can be tuned via exposure to air. By modulating the doping level, the author reported a relative high figure-of-merit ZT of 0.14. Even though this work gave detail characterization on the doped film and the potential application of perovskites, it's still need reconsideration after dealing with the problems below.

- (1) As referred by the author, there are some reports on the thermoelectric performance of halide perovskite material (PNAS, DOI: 10.1073/pnas.1711744114). The authors claim that previous study is

lack of in-depth understanding of physical mechanisms and the strategies for identification of ZT is not clear. However, the author defined this work as “enhanced control” for “improved” TE performance. I don't think a clear description on the deep analysis and mechanism are well summarized in the manuscript.

(2) The author demonstrated they obtained “improved” TE performance. However, the literature based on the halide perovskite material shown a ZT of 0.11 at 320 K (PNAS, 2017, DOI: 10.1073/pnas.1711744114). More recently, Saini et al. demonstrated a high ZT of 0.137 with spin-coated CsSnI₃ films at room temperature. Hence, the TE performance of CsSnI₃-xCl_x perovskites here is not impressive compared with previous reports. Of particular note, the high ZT of 0.137 has been obtained for the halide perovskites via a simple thermal annealing treatment. Is it possible for the authors to further improve the thermoelectric performance in this case? Related experiments and/or comments should be provided.

(3) Stability is a key issue for both thermoelectric materials and perovskite material. As mentioned in the manuscript, the electrical conductivity and Seebeck coefficient of materials are sensitive to the air exposure time (Figure 4e). It seems that the thermoelectric performance of the perovskite is not stable upon air exposure and high temperature environment. The author should characterize the long-term stability and the temperature-dependent stability of the materials.

(4) As investigated in this work, the effective oxidation thickness is ~7.5 nm for the bulk film. What's the thickness used here for calculating the electric conductivity? The author explained that the surface oxidation played the critical role for tuning the electric conductivity of the mixed CsSnI₃-xCl_x perovskite thin films. But how can the author confirm the holes transport properties in the bulk film with different phases? Moreover, the thermal conductivity is also affected by the electron transport properties, phonon diffraction, edge effect, etc. It's better to study TE properties of thinner films and make a direct comparison with the value of thick films.

(5) What's the humidity controlled in this study? The conductivity was modulated with oxidation exposure to air, which should be affected by both the water, and O₂ in the atmosphere.

(6) What's the seed layer effects on the morphology, the conductivity and thermoelectric properties of the halide CsSnI₃-xCl_x perovskite?

(7) The data shown in this manuscript don't have error bar. The author to provide the repeatability, uniformity and reliability data of devices performance.

Reviewer #3 (Remarks to the Author):

Liu et al. reported the thermoelectric properties of all-inorganic CsSnI₃-xCl_x perovskites with enhanced air stability. They fabricated perovskite films via three kinds of methods and observed tuning of the thermoelectric properties by self-doping through the oxidation of Sn²⁺ to Sn⁴⁺ in a thin surface-layer. They claimed that this strategy can separate the doping defects from the transport region, enabling enhanced electrical conductivity. They also used chlorine-rich surface layer to protect the bulk film from oxidation. In the end they achieved a ZT of 0.14 with optimized chlorine-doping and degree of the oxidation. The work is generally interesting, but there are problems with it, since the choice of a material will eventually be deteriorated is not wise. Therefore, what we observed here are just a flashing transient state of the material. This fact is in the opposite of the fame of thermoelectric device as stable and reliable power source. Therefore, I will not suggest the publication of the work in Nature Communications.

There are detailed concerns below:

1. What is humidity of the air, which the samples were expected to? As far as I know, the humidity may also affect the perovskite materials dramatically.
2. In the case of Sn(II) based perovskite, light will also play a role in the doping process. How will this affect the conductivity?

3. What is the carrier concentration of the samples before and after the air exposure? According to the discussion, the carrier concentration should increase after air exposure.
4. What is the mobility of the samples before and after the air exposure?
5. When the author measures the electrical properties of sample under nitrogen atmosphere, the electrical conductivity also cannot stay at a constant as a function of time. The oxidation of Sn^{2+} to Sn^{4+} may be also caused by the interaction between Sn ions and halide ions or other aspects. Without comparison, it is not very convincing to conclude that air exposure is the only reason for the oxidation of Sn^{2+} to Sn^{4+} .
6. Ref 4 should be updated, since the highest PCE is above 24%.
7. On page 4, the claim of doping Cl in the top 10 nm layer is difficult, since it will change with time by ion migration.
8. The exists of Cl layer will only slow down the oxidation as mentioned by the author, it can not solve the problem.
9. On page 6, the reason for the different conductivity change behavior is unclear and should be explained or supposed.
10. Page 9, the Sn^{4+} on the surface will protect the bottom from oxidation is hard to understand. It is not the case of Al_2O_3 on Al. This is opposite to the observation in page 10.

Point-by-point response to reviewers' comments:

Reviewer #1 (Remarks to the Author):

General comment_R1: In the present work, Fenwick et al. explore all-inorganic perovskite CsPbI₃ for thermoelectric application by self-doping and addition of chlorine to further enhance the stability and performance. A ZT of 0.14 was recorded which is claimed to be the highest for halide-perovskite thermoelectrics. While this is one of the most detailed work on halide perovskite thermoelectrics, there is still a long way before appreciable ZT and stability can be achieved for halide perovskite thermoelectrics. Below are some detailed comments for a major revision.

Reply to general comment_R1: We thank the referee for valuable comments on our work and reply to the constructive questions in detail below.

Comment_R1.1: Thermoelectric performance of CsSnI₃ in the form of nanowire as well as the self-doping process in Sn-based perovskites are already reported as mentioned in the manuscript. Another major reservation about the present work is a recent report (<https://doi.org/10.1557/adv.2019.279>) on CsSnI₃ thin films where a ZT of 0.137 was achieved which is equal to the present work after error estimations without any sample modification. Authors need to compare their work with prior reports and highlight the unique aspects in a separate paragraph.

Author reply R1.1: As pointed out, Saini et al. reported a ZT of 0.137 with spin-coated CsSnI₃ thin films [MRS Advances, DOI: 10.1557/adv.2019.279]. In fact, this result came out after we submitted our manuscript, which is why we hadn't referred to it in the text. It was published as a very brief proceeding from the MRS Spring Meeting 2019 with few details, and in their conclusions the authors describe their own work as "preliminary results". Although they do achieve a decent ZT value, they study the effect of baking temperature and see huge variations in performance but fail to observe any trends. This seems to indicate that they don't have a good control over the synthesis, morphology or

doping conditions. We do exert control on all of these parameter and observe clear trends, which allows us to understand the materials more deeply. Consequently, our work goes much deeper into the mechanisms of doping and how this affects thermoelectric performance, which is important for advancing the field of perovskite thermoelectrics, but also raises some interesting results for the halide perovskite community a whole.

There are some key differences between our work and Saini et al.'s. Firstly, in Saini's work, they deposited films with spin-coating, whereas we have developed vapour deposition processes. From their XRD patterns, the spin-coated films have (202) orientation. The films have cracks and pin holes preventing any meaningful trends between treatments from being observed. In our work, our films have different dominant orientations depending on deposition process, such as (202) for sequential method and (220) for co-evaporation method. Our films are pin hole free (SEM images) and mirror black (optical images). This allowed us to probe the relationships between grain orientation, morphology, stability and conductivity that turn out to be important.

Secondly, in Saini's work, the measurement of electrical conductivity and Seebeck coefficient was in-plane. Thermal conductivity measurement was out of plane, which is widely known as being a major source of error in reported ZT values of thin films. In our manuscript, all three parameters were measured in-plane.

Most importantly, it is well known that CsSnI_3 has a self-doping process with Sn^{2+} oxidizing to Sn^{4+} . The oxidation process plays a key role in thermoelectric properties such as electrical conductivity and Seebeck coefficient. In Saini's work, they probed the effect of baking temperature on ZT values. There was no discussion of the oxidation nor of degradation process during the measurement, nor of the exposure time to air. In our work, we investigated the thermoelectric properties as a function of these self-doping processes. As a result of this, we establish a mixed halide approach to improve stability and enhance control over doping process. These mixed halide perovskite films with improved stability were used to analyse the self-doping process for thermoelectrics.

Our action R1.1: We have removed the claim that Lee et al. had the previous highest ZT in the perovskite class of materials and have replaced this with the ZT of Saini et al.. We have also reworded a statement implying that our ZT value is exceptional. The affected sentences now read:

(Page 4) “In 2017, Lee *et al.* reported the ultralow thermal conductivity of a single CsSnI₃ nanowire and ZT ~ 0.11 at 320K,²² whilst Saini et al. report a ZT in thin films of 0.137.”

(Page 5) “...and achieve a remarkably high ZT for halide perovskites of 0.14 at 345K upon simultaneous optimisation of the degree of Cl-doping and the degree of oxidation.”

(Page 11) “This is the highest ZT recorded for a halide perovskite and was achieved with a continuous film rather than a single nanocrystal.”

Comment_R1.2: Why different degree of oxidation and degradation was observed for the films deposited by different techniques even though the composition was same? Does orientation plays an important role in conductivity enhancement and degradation in air?

Author reply R1.2: Our results show that film morphology plays a key role in stability of films and the potential for high conductivity. Different degrees of oxidation and degradation were observed in the films made by the three deposition methods. These three types of films have different dominant orientations and, previously, grain orientation has been shown to have a large impact on degradation rates in air [Ma et al., *Nanoscale*, **2019**, 11, 170]. In our work, co-evaporated films with dominant (220) orientation show a good air stability but low electrical conductivity. Sequentially deposited films with dominant (202) orientation show a poor air stability but high electrical conductivity. SLS films combining co-evaporation (seed layer) with sequential deposition have a mixed orientation of (220) and (202). These films show a comparable electrical conductivity to sequentially deposited films and

intermediate air stability. However, morphology is an interplay between grain size, grain orientation and topography, and we cannot isolate the effect of only grain orientation experimentally.

Our action R1.2: We have added discussion of this point on page 6, and now it reads: “Co-evaporated films with dominant (220) orientation therefore show the best air stability but lowest electrical conductivity. Sequentially deposited films with dominant (202) orientation show a poor air stability despite their larger grain sizes, but do have higher electrical conductivity. Previous work has shown that grain orientation can have a significant effect on degradation rates of halide perovskite films [Ma et al., *Nanoscale*, 2019, 11, 170] and this is likely to be the case here.”

Comment_R1.3: On page 6, authors mention that films deposited by SLS method shows the best stability and that is why they chose this sample for further investigation. But if we look at Figure 1-i, it appears that conductivity of co-evaporated film is more stable after 20 minutes of air exposure. In addition, the morphology of co-evaporated films are significantly better than other films.

Author reply R1.3: On page 6, we mentioned that the films deposited by SLS have better stability than those deposited sequentially (“SLS films have enhanced stability compared to the sequentially deposited films.”). However, we do not claim that they have an enhanced stability compared to the coevaporated films, as this is clearly not the case. The reason for not selecting coevaporated films for further investigation is that they only achieve a maximum electrical conductivity of 1.2 S/cm after optimal oxidation, whereas SLS and co-evaporated films achieve maximum electrical conductivities of 37.1 S/cm and 32.8 S/cm respectively. We consider 1.2 S/cm too low for thermoelectric applications, and therefore rejected co-evaporated films from the thermoelectric study despite higher stability and smoother film formation. Although the coevaporated films have the smoother surface topography, all three films show a high quality perovskite structure in the XRD and continuous films, so we see no morphological reason to avoid the SLS films. The stability of SLS films is later improved by Cl-doping with no negative effect on the conductivity.

Action R1_3: To make this reasoning clearer in the manuscript, we have re-worded a section on page 6. It now reads:

“CsSnI₃ thin films deposited by SLS show a similar maximum electrical conductivity (37.1 S/cm) to sequentially deposited films (32.2 S/cm), which is ~25 times higher than the maximum for coevaporated films (1.2 S/cm), a value we consider too low for thermoelectric applications.”... “Since SLS films have enhanced stability compared to the sequentially deposited films, we chose SLS produced films as the platform from which to optimise the thermoelectric properties of CsSnI₃ perovskites. “

Comment_R1.4: Authors claim that the surface doping technique and inclusion of Cl enhances the stability of CsSnI₃. So it is imperative that performance stability of the best sample measured under nitrogen and air over the course of few hours/days be included.

Author reply R1.4: We appreciate this comment, and to address it we performed stability measurements in air and in nitrogen. As shown in Figure R1 (below), the figure of merit (ZT) value for unencapsulated CsSnI_{3-x}Cl_x films decreased 32% after 10 hours in air (Figure R1, left), which is consistent with our previous UV-vis stability measurements (Figure S8 in original submission). When the films were stored in nitrogen between measurements, the ZT value kept 70% of its initial value after 10 days (Figure R1, right). We also need to highlight that each time we measured the sample stored in nitrogen it came into contact with air for approximately 2 minutes during transfer between the glovebox and the instrument. When we consider the cumulative air exposure in Figure R1 (nitrogen) (which totals ~18 minutes) with Figure 4(d) of the manuscript, we see a similar rate of degradation, indicating that the degradation is dominated by the air exposure between measurements and not by the period of storage in nitrogen. We conclude that when samples are chlorine doped and kept away from air, they are surprisingly stable.

Figure R1. Performance stability measurement of 1% Cl doping CsSnI_{3-x}Cl_x perovskite films without encapsulation in air with 40% humidity (left) and in nitrogen (right). It should be noted that samples stored in nitrogen are exposed to air for up to 2 minutes between each measurement due to the transfer time from our instrument to the glovebox and therefore the righthand panel encompasses 18 minutes of air exposure and ~10 days of nitrogen storage.

Our action R1.4: We have added the performance stability measurements as a new section in the Supporting Information (Supporting Information section 12). [not repeated here]

In the main text (page 11) we have included the following statement:

“The figure of merit shows a 32% reduction after 10 hours in air, and a 30% reduction after 10 days storage in nitrogen atmosphere (Supporting Information S17-S18).”

Comment_R1.5: Authors need to report errors in their measurements for Seebeck and thermal conductivity. Since authors mentioned that the thickness of the films were between 250-300 nm, this range will also contribute to the error estimations of different parameters. This will induce an error in the final value of ZT which should be reported.

Author reply R1.5: We have added the errors for electrical and thermal conductivity and Seebeck coefficient in the revised manuscript. The errors of electrical and thermal conductivity are dominated by the measurement of film thickness as the referee has suggested. The errors on the Seebeck

coefficient come from the fitting error of thermal voltage vs. temperature gradient data. Because the film thicknesses used in electrical and thermal conductivity measurements are identical, they cancel out in the calculation of ZT , limiting the error on the final value. In fact, this is the intrinsic advantage of measuring Seebeck coefficient, electrical and thermal conductivities on the same sample.

Our action R1.5: We have added an explanation of the main sources of error in the Supporting Information (section 11). This reads:

“The errors of electrical and thermal conductivity are dominated by the measurement of film thickness. The errors on the Seebeck coefficient come from the fitting error of thermal voltage vs. temperature gradient data. Because the film thicknesses used in electrical and thermal conductivity are identical, they cancel out in the calculation of ZT , limiting the error on the final value.”

We have added errors in Figure 4 (a-f) as shown in the revised manuscript. Note that because of the small error on the Seebeck coefficient, error bars are not visible in Figure 4b (For example at 292K the Seebeck coefficient is $144.69 \pm 1.53 \mu\text{V/K}$). This is explained in the revised caption. Moreover, we have added error ranges to values discussed throughout the main text.

Comment_R1.6: What is the carrier density before and after doping since it is an important parameter in the present work.

Author reply R1.6: This is a very fair comment and we have now performed Hall effect measurements on 1% Cl doped $\text{CsSnI}_{3-x}\text{Cl}_x$ perovskite thin films. As we can see in Figure R2 (below), before doping, the initial charge carrier density was $2.38 \times 10^{18} \text{ cm}^{-3}$, and after air exposure, the charge density increased to $1.06 \times 10^{19} \text{ cm}^{-3}$. This indicates that our doping method based on the oxidation of Sn^{2+} to Sn^{4+} does achieve charge carrier densities suitable for thermoelectrics. Meanwhile, the Hall mobility decreased with air exposure from $76.1 \text{ cm}^2 \text{ V}^{-1} \text{ s}^{-1}$ to $50.1 \text{ cm}^2 \text{ V}^{-1} \text{ s}^{-1}$ after doping, because the defects introduced from doping act as scattering centers for charge carriers. The opposing effects of increasing charge carrier concentration and decreasing mobility are the reason that a maximum in electrical conductivity is reached after modest air exposure. The maximum Hall mobility of $76.1 \text{ cm}^2 \text{ V}^{-1} \text{ s}^{-1}$ is

smaller than the value of $394 \text{ cm}^2 \text{ V}^{-1} \text{ s}^{-1}$ reported in CsSnI_3 single crystals. [Lee et al., DOI:10.1073/pnas.1711744114]

Our action R1.6: We have added Hall measurement details in supporting information section 15 (not repeated here), and a summary of this in the methods section of the main text (not repeated here). We have also replaced Figure 4e with Figure R2 (below) in the main text.

We have added some discussion of the result on page 10 of the main text: “To verify this hypothesis, we used Hall measurements to determine the charge carrier concentration as a function of air exposure, showing an increase with air exposure from $2.38 \times 10^{18} \text{ cm}^{-3}$ to $1.06 \times 10^{19} \text{ cm}^{-3}$ after 12 minutes. Meanwhile, the Hall mobility decreases from an initial value of $76.1 \text{ cm}^2 \text{ V}^{-1} \text{ s}^{-1}$ to $50.1 \text{ cm}^2 \text{ V}^{-1} \text{ s}^{-1}$ after oxidation.”

Figure R2. Charge carrier density and Hall mobility of $\text{CsSnI}_{3-x}\text{Cl}_x$ perovskite thin films (1% Cl-doped) plotted as a function of air exposure time.

Comment_R1.7: Authors need to give more details of how they extracted the Lorentz number. Further, for Figure S11, authors have plotted multiple data points of thermal conductivity vs electrical conductivity for a single temperature. How different values of electrical and thermal conductivity were achieved at a single temperature? Is this achieved by different degree of oxidation of the sample? More details should be mentioned.

Author reply R1.7: Thanks for pointing out this issue. As we mentioned in supporting information, Lorenz number was extracted from the equation $\kappa = \kappa_{lattice} + \sigma LT$. As the reviewer suggests, we obtained data of different electrical conductivities at the same temperature by exploiting different degrees of oxidation on the same sample. The thermal and electrical conductivities of the sample were measured from 20°C to 80°C then the sample was exposed to air for 3 minutes, before being measured again over the same temperature range (σ_{3mins} and κ_{3mins}). Then the same sample was exposed to air for an additional 3 minutes. Thus, we obtained several values (σ_{time} and κ_{time}) in the same sample and plot κ_{time} vs. σ_{time} . The slope of this plot gives the Lorenz number times temperature ($L*T$) and the intercept is lattice thermal conductivity ($\kappa_{lattice}$).

Our action R1.7: We have now added more details about Lorenz number extraction in Supporting Information section 9.

"To do this, after the first measurement of the sample, it was exposed to air for 3 minutes before re-measuring the electrical and thermal conductivity (σ_{3mins} and κ_{3mins}). Then the same sample was exposed to air for an additional 3 minutes, before measuring the electrical and thermal conductivity again (σ_{6mins} and κ_{6mins}). Thus, we obtained several values (σ_{time} and κ_{time}) and plot σ_{time} vs. κ_{time} . From the equation, $\kappa_{time} = \kappa_{lattice} + \sigma_{time}LT$, we see that the slope gives Lorenz number times temperature ($L*T$) and the intercept is lattice thermal conductivity."

Reviewer #2 (Remarks to the Author):

General comment_R2: In this manuscript, Liu T. et al. reported a study on the thermoelectric properties of vacuum deposited CsSnI_{3-x}Cl_x perovskites via different procedures. With the surface oxidation of Sn²⁺ to Sn⁴⁺, the electrical conductivity of the bulk film can be tuned via exposure to air. By modulating the doping level, the author reported a relative high figure-of-merit ZT of 0.14. Even though this work gave detail characterization on the doped film and the potential application of perovskites, it's still need reconsideration after dealing with the problems below.

Reply to general comment_R2: We thank the referee for reviewing the manuscript and respond to the specific comments below.

Comment_R2.1: As referred by the author, there are some reports on the thermoelectric performance of halide perovskite material (PNAS, DOI: 10.1073/pnas.1711744114). The authors claim that previous study is lack of in-depth understanding of physical mechanisms and the strategies for identification of ZT is not clear. However, the author defined this work as “enhanced control” for “improved” TE performance. I don't think a clear description on the deep analysis and mechanism are well summarized in the manuscript.

Author reply R2.1: Thanks for the suggestion. Lee et al. (PNAS, DOI:10.1073/pnas.1711744114) reported the discovery of ultralow thermal conductivity of single crystalline CsSnI₃ perovskites with high electrical conductivity (282 S/cm) and high hole mobility (394 cm²V⁻¹s⁻¹). Their work gave insights into low thermal conductivity all-inorganic perovskite thermoelectric materials, studying three systems (CsPbI₃, CsPbBr₃ and CsSnI₃) which was interesting and useful for the community. Their work concentrated on the explaining the reason for ultralow thermal conductivity, where a cluster rattling mechanism dominated. They then further investigated the transport properties of CsSnI₃ nanowires, reporting the power factor and ZT, but without any attempt at optimization. In our work, we investigated the doping process by air oxidation in Sn based perovskites films. Furthermore, we introduced Cl into

CsSnI₃ perovskites to form a mixed halide perovskite (CsSnI_{3-x}Cl_x) and improving air stability. In this case, we could fine tune the electrical conductivity by self-doping through air oxidation. The optimized figure of merit is 10 times higher than the original value before intentional doping, and is marginally higher than the value in single crystals reported by Lee et al.. In the revised manuscript, we also have more information on the evolution of charge carrier concentration and mobility with oxidation (Hall measurements), and detailed stability measurements. We therefore consider this work to be original and important.

Our action R2.1: To address this comment, additional experiments and analysis have been included in the revised text and supporting information. Hall effect measurements were performed to investigate the charge carrier density and mobility before and after doping by air oxidation, which reveals details of the doping mechanism in Sn based perovskites. (Details of this are outlined in Reply 1.6, and have been summarized in the revised Supporting Information section 15) Long term stability measurements of thermoelectric properties (ZT) in air and in nitrogen atmosphere have been investigated for the improved stability description. (Details outlined in reviewer response R1.4 and in the revised Supporting Information section 12)

Comment_R2.2: The author demonstrated they obtained “improved” TE performance. However, the literature based on the halide perovskite material shown a ZT of 0.11 at 320 K (PNAS, 2017, DOI: 10.1073/pnas.1711744114). More recently, Saini et al. demonstrated a high ZT of 0.137 with spin-coated CsSnI₃ films at room temperature. Hence, the TE performance of CsSnI_{3-x}Cl_x perovskites here is not impressive compared with previous reports. Of particular note, the high ZT of 0.137 has been obtained for the halide perovskites via a simple thermal annealing treatment. Is it possible for the authors to further improve the thermoelectric performance in this case? Related experiments and/or comments should be provided.

Author reply R2.2: As pointed out by the referee, Saini et al. reported a ZT of 0.137 with spin-coated CsSnI₃ thin films [MRS Advances, DOI: 10.1557/adv.2019.279]. In fact, this result came out after we submitted our manuscript, which is why we hadn't referred to it in the text. It was published as a very brief proceeding from the MRS Spring Meeting 2019 with few details, and in their conclusions the authors describe their own work as "preliminary results". Although they do achieve a decent ZT value, they study the effect of baking temperature and see huge variations in performance but fail to observe any trends. This seems to indicate that they don't have a good control over the synthesis, morphology or doping conditions. We do exert control on all of these parameter and observe trends, which allows us to understand the materials more deeply. Consequently, our work goes much deeper into the mechanisms of doping and how this affects thermoelectric performance, which is important for advancing the field of perovskite thermoelectrics, but also raises some interesting results for the halide perovskite community a whole.

In the response to reviewer 1 (response R1.1), we list further details of the unique aspects in our work. Most importantly, this paper is about understanding the doping and optimization process in tin perovskites for thermoelectrics, not simply about reporting a high ZT. On the specific point of thermal annealing, we note that the thermal annealing in Saini's work was necessary to form a continuous perovskite film after spin-coating the precursor solution. In our work, baking at 170 °C was also used to form the CsSnI₃ perovskite films when depositing the precursors sequentially and by the SLS method. However, our study then went much deeper, using a dual optimization of oxidation and chlorine doping to obtain a maximum ZT, delivering an understanding of the doping processes to achieve this as well as a method to improve stability. In fact, since the work of Saini et al. does not control the air exposure time of their samples, it is no surprise that there is no trend in their "preliminary results". Our revised manuscript goes even more deeply into these mechanisms, as we have now provided Hall measurements and long term stability measurements.

Comment_R2.3: Stability is a key issue for both thermoelectric materials and perovskite material. As mentioned in the manuscript, the electrical conductivity and Seebeck coefficient of materials

are sensitive to the air exposure time (Figure 4e). It seems that the thermoelectric performance of the perovskite is not stable upon air exposure and high temperature environment. The author should characterize the long-term stability and the temperature-dependent stability of the materials.

Author reply R2.3: This is a very reasonable comment, and to address it, we performed stability measurements in nitrogen and in air, that are incorporated in the Supporting Information section 12. As shown in Figure R1, the figure of merit (ZT) value for unencapsulated films ($\text{CsSnI}_{3-x}\text{Cl}_x$) decreased 32% after 10 hours in air, which is consistent with our stability measurements from the UV-vis results (Figure S8 in original submission). When the films were stored in nitrogen between measurements, the ZT value kept 70% of its initial value after 10 days. In fact each time we measured the sample stored in nitrogen it came into contact with air for up to 2 minutes during transfer between the glovebox and the instrument. When we consider Figure R1 (nitrogen) in terms of cumulative air exposure (18 minutes in total) and compare this with Figure 4(d) of the manuscript, we see a similar rate of degradation, indicating that the degradation is dominated by the air exposure between measurements and not by the period of storage in nitrogen. We conclude that when samples are chlorine-doped and kept away from air, they are remarkably stable.

Figure R1. Performance stability measurement of 1% Cl doping $\text{CsSnI}_{3-x}\text{Cl}_x$ perovskite films without encapsulation in air with 40% humidity (left) and in nitrogen (right). It should be noted that samples stored in nitrogen are exposed to air for up to 2 minutes between each measurement due to the transfer time from our instrument to the glovebox and therefore the righthand panel encompasses 18 minutes of air exposure and ~10 days of nitrogen storage.

Our action R2.3: We have added the performance stability measurements as a new section in the Supporting Information (Supporting Information section 12). [not repeated here]

In the main text (page 11) we have included the following statement: “The figure of merit shows a 32% reduction after 10 hours in air, and a 30% reduction after 10 days storage in nitrogen atmosphere (Supporting Information S17-S18).”

Comment_R2.4: As investigated in this work, the effective oxidation thickness is ~7.5 nm for the bulk film. What’s the thickness used here for calculating the electric conductivity? The author explained that the surface oxidation played the critical role for tuning the electric conductivity of the mixed CsSnI_{3-x}Cl_x perovskite thin films. But how can the author confirm the holes transport properties in the bulk film with different phases? Moreover, the thermal conductivity is also affected by the electron transport properties, phonon diffraction, edge effect, etc. It’s better to study TE properties of thinner films and make a direct comparison with the value of thick films.

Author reply R2.4: The thickness of the oxidation layer is ~7.5nm (as identified by the referee), and the full thickness of the films are 250 – 300 nm. The oxidized layer therefore represents only 2-3% of the full film thickness. For calculating the electrical and thermal conductivity, we use the full film thickness (250 – 300 nm). We therefore considered the surface oxidized layer and bulk layer as a whole and investigated the transport properties based on the whole film. Since the oxidized layer is such a small proportion of the full film thickness (2-3%), we can be sure that the measured properties represent those of the underlying film.

To address whether thermal conductivity is affected by the electron transport properties, phonon diffraction (scattering), edge effects etc, we have additionally performed experiments on thinner films – as suggested by the referee. These films are of 115 ± 5 nm thick and have are presented in Figure S21 of the Supporting Information (section 14). The electrical conductivity at room temperature is 168.6 ± 7.2 S/cm for the sample exposed to air for 9 minutes, which is higher than the highest value in the film

with thickness 265 ± 15 nm (126.5 S/cm, air exposure 9 minutes). At longer air exposure times, the conductivity of the thinner film increases to a maximum of ~ 200 S/cm⁻¹, which is significantly higher than any value observed for the thicker films. However, the initial electrical conductivity in thinner films measured in glovebox with no air exposure is 1.8 S/cm, which is lower than the value in thick films (8.6 S/cm). This trend can be explained in the following way. Nanostructuring plays a key role in thermoelectric properties, through mechanisms such as grain boundaries, edge effect and point defects. In this case, the thinner films with increased grain boundaries and surface scattering will result in lower hole mobility than that in thick films, which is consistent with our initial measurements of electrical conductivity before oxidation. When the films are exposed to air, the oxidation process of Sn²⁺ to Sn⁴⁺ will initiate in the surface layer. That oxidation process will also happen from the grain boundaries, which are more numerous in the thinner films. With the increasing charge concentration, the transport energy level will shift closer to the Fermi level, resulting in a reduced Seebeck coefficient. [Lu et al., PCCP, 2016, 18, 19503-19525] [Germs et al., PRL 109, 016601 (2012)] Consequently, the Seebeck coefficient of the thin films is in the range 60.6 to 113.3 μ V/K, which is smaller than in thick films (103.0 to 144.7 μ V/K). This is shown in In Figure S21 (b), above.

As a consequence of the higher electrical conductivity in the thin films, the thermal conductivity (Figure S21 (c)) is also higher than in thick films, attributed to the larger electron contribution to thermal transport.

Overall, the highest figure of merit in thinner films is 0.09 ± 0.004 at 350 K which is lower than the maximum of 0.14 measured for the thicker films. We can therefore say that although the different morphology of the thin films is conducive to better electrical conductivity, this does not translate into an improved thermoelectric figure of merit, ZT .

Our action R2.4:

Figure S21 Thermoelectric properties of 1% Cl doping CsSnI_{3-x}Cl_x films with thickness of 115 ± 5 nm. (a) Electrical conductivity, (b) Seebeck coefficient, (c) thermal conductivity and (d) figure of merit.

We have added Figure S21 (above) in the Supporting Information and associated discussions in Supporting Information section 14: “We performed thermoelectric property measurements of thinner films of mixed halide perovskites with thickness of 115 ± 5 nm. The electrical conductivity at room temperature is 168.6 ± 7.2 S/cm for the sample exposed to air for 9 minutes, which is higher than the highest value in the film with thickness 265 ± 15 nm (126.5 S/cm, air exposure 9 minutes). At longer air exposure times, the conductivity increases to ~200 Scm⁻¹ in the thinner films, which is significantly higher than any value observed for the thicker films. However, the initial electrical conductivity in thinner films measured in glovebox with no air exposure is 1.8 S/cm, which is lower compared to the value in thick films (8.6 S/cm). This trend can be explained in the following way. Nanostructuring plays a key role in thermoelectric properties, through mechanisms such as grain boundaries, edge effect and point defects. In this case, the thinner films with increased grain boundaries and surface scattering will result in lower hole mobilities than in thick films, which is consistent with our initial measurements of electrical conductivity before oxidation. When the films are exposed to air, the oxidation of Sn²⁺ to Sn⁴⁺ will initiate in the surface layer. That oxidation process will also happen from the grain boundaries,

which are more numerous in the thinner films. With the increasing charge concentration, the transport energy level will shift closer to the Fermi level, resulting in a reduced Seebeck coefficient. [Lu et al., PCCP, 2016, 18, 19503-19525] [Germs et al., PRL 109, 016601 (2012)] Consequently, the Seebeck coefficient of the thin films is in the range 60.6 to 113.3 $\mu\text{V/K}$, which is smaller than in thick films (103.0 to 144.7 $\mu\text{V/K}$). This is shown in In Figure S21 (b), above. As a consequence of the higher electrical conductivity in the thin films, the thermal conductivity (Figure S21 (c)) is also higher than in thick films, attributed to the larger electron contribution to thermal transport.

Overall, the highest figure of merit in thinner films is 0.09 ± 0.004 at 350 K, which is lower than the maximum of 0.14 measured for the thicker films. We can therefore say that although the different morphology of the thin films is conducive to better electrical conductivity, this does not translate into improved thermoelectric figure of merit, ZT .”

In the main text we have added a sentence on page 11:

“We also note that thinner films showed higher electrical conductivities, but no improvement in ZT (Supporting Information section 14).”

Comment_R2.5: What’s the humidity controlled in this study? The conductivity was modulated with oxidation exposure to air, which should be affected by both the water, and O₂ in the atmosphere.

Author reply R2.5: Thanks for the comment. The referee is correct that the humidity of the air will affect the oxidation time. We have measured the humidity in our lab, and it is ~40%. The O₂ levels are, of course, constant. This is important to note in the manuscript, which we now do.

Our action R2.5:

We have reported the humidity of our lab in the methods section of the main manuscript.

“Samples fabricated in the glovebox were transferred to the Linseis Thin Film Analyser with < 2 minutes exposure to air. The humidity in the lab was ~40%.”

Comment_R2.6: What's the seed layer effects on the morphology, the conductivity and thermoelectric properties of the halide CsSnI_{3-x}Cl_x perovskite?

Author reply R2.6: The seed layer was deposited by co-evaporating SnI₂ and CsI. Coevaporated layers have a grain orientation of (220) (Figure 1g). Sequentially deposited perovskite layers have a dominant grain orientation of (202) (Figure 1g). In the case of the SLS method, the coevaporated seed layer provides a (220) surface for the sequentially deposited film on top, resulting in a mixed grain orientation film overall.

As we can see from Figure 1 in main text, the coevaporated films with dominant (220) grain orientation have longer air stability but lower electrical conductivity than the films with dominant orientation of (202) deposited by the sequential method. With the seed layer method, we can get intermediate air stability and high electrical conductivity. Chlorine doping of this structure yields the highest stability and electrical conductivity. The seed layer effect on the morphology was discussed by SEM in main text Figure 1. The effect on the grain orientation was further discussed by GIWAX in Supporting Information Figure S2.

Comment_R2.7: The data shown in this manuscript don't have error bar. The author to provide the repeatability, uniformity and reliability data of devices performance.

Author reply R2.7: Thanks for pointing out this issue. This was also raised by reviewer 1, and we have responded in **reply R1.5**. In summary, we have added error bars in Figure 4 (a), (b), (c), (d) and (f). The main component of the error bar of electrical and thermal conductivity comes from the film thickness. The main contribution to the error bar of Seebeck coefficient comes from the fitting error of thermal voltage vs. temperature gradient. We have also added error bars to values discussed in the main text. To address the comment that the repeatability, uniformity and reliability data of device performance, we also added a distribution figure of devices performance of 1% Cl doping CsSnI_{3-x}Cl_x films in Supporting Information, Figure S13.

Our action R2.7: The actions on error bars are listed in “**Our actions R1.5**”. We have also added a histogram of ZT values for multiple 1% Cl doped $\text{CsSnI}_{3-x}\text{Cl}_x$ films in the Supporting Information Section 13.

Reviewer #3 (Remarks to the Author):

General comment_R3: Liu et al. reported the thermoelectric properties of all-inorganic $\text{CsSnI}_{3-x}\text{Cl}_x$ perovskites with enhanced air stability. They fabricated perovskite films via three kinds of methods and observed tuning of the thermoelectric properties by self-doping through the oxidation of Sn $2+$ to Sn $4+$ in a thin surface-layer. They claimed that this strategy can separate the doping defects from the transport region, enabling enhanced electrical conductivity. They also used chlorine-rich surface layer to protect the bulk film from oxidation. In the end they achieved a ZT of 0.14 with optimized chlorine-doping and degree of the oxidation. The work is generally interesting, but there are problems with it, since the choice of a material will eventually be deteriorated is not wise. Therefore, what we observed here are just a flashing transient state of the material. This fact is in the opposite of the fame of thermoelectric device as stable and reliable power source. Therefore, I will not suggest the publication of the work in Nature Communications.

Reply to general comment_R3: We appreciate the review from the referee and we are glad that they are found to be interesting. In our response we have taken a much closer look at the stability of the materials with additional experiments. Many of the specific comments of this referee are concentrated on ion migration and optoelectronic properties. We have now confirmed that ion migration and light exposure do not affect the thermoelectric properties. This was done through extensive additional experiments based on perovskite solar cells and photodetectors made of $\text{CsSnI}_{3-x}\text{Cl}_x$. We believe that the specific comments of this referee (below) have been fully addressed.

Comment_R3.1: What is humidity of the air, which the samples were expected to? As far as I know, the humidity may also affect the perovskite materials dramatically.

Author reply R3.1: We have measured the humidity in our lab, and it is ~40%. We note that the samples were prepared in a nitrogen glovebox and transferred to the measurement equipment, being exposed to the air for < 2 minutes before the first measurement. All measurements were performed in vacuum. When we wished to partially oxidise the films, the measurement chamber was refilled with air to atmospheric pressure for a designated time before pumping down again for the next measurement.

Our action R3.1:

We have reported the humidity of our lab in the methods section of the main manuscript.

“Samples fabricated in the glovebox were transferred to the Linseis Thin Film Analyser with < 2 minutes exposure to air. The humidity in the lab was ~40%.”

Comment_R3.2: In the case of Sn(II) based perovskite, light will also play a role in the doping process. How will this affect the conductivity?

Author reply R3.2: Thanks for pointing this out. Firstly, we should point out that all thermoelectric measurements were performed in the dark in a vacuum chamber with no optical windows. To address the comment on the effect of light on doping in Sn-based perovskites, we performed electrical conductivity measurements of $\text{CsSnI}_{3-x}\text{Cl}_x$ perovskite films both in dark and under AM1.5G illumination. To do this, a photodetector was fabricated by depositing $\text{CsSnI}_{3-x}\text{Cl}_x$ films on a silicon oxide surface (silicon substrate with an insulating thermal oxide layer). 80 nm Au was deposited through a shadow mask to form top contacts. Current voltage curves were measured using a Keithley 2400 sourcemeter, and AM 1.5G illumination from a solar simulator was used as light source. One sample was encapsulated in a nitrogen atmosphere before measurement, whilst the other was exposed to air for 20 minutes prior to measurement.

Figure R3. I-V measurement of photodetector in the dark and under AM1.5G illumination. Left: encapsulated sample without prior air exposure. Right: sample kept in ambient air for 20 minutes before measurement.

As shown in Figure R3 (above), there was a light response in encapsulated samples without prior air exposure. Electrical conductivity changed from 0.8 S/cm in dark to 10.5 S/cm upon simulated solar illumination. However, for samples with 20 minutes air exposure prior to the measurement, the current was identical in the dark and under illumination at a higher value of 82.5 S/cm. Charge carriers introduced by solar irradiation were therefore deemed to be of negligible effect in high conductivity CsSnI_{3-x}Cl_x semiconductors.

Our action R3.2:

We have made an addition to the methods section of the main text to highlight that we performed measurements in the dark.

“All measurements were performed under vacuum and in the dark.”

We have also included Figure R3 and discussion of the associated methods in the Supporting Information (Section 16 “Photoconductivity in CsSnI_{3-x}Cl_x thin films”). [discussion not repeated here]

Comment_R3.3: What is the carrier concentration of the samples before and after the air exposure? According to the discussion, the carrier concentration should increase after air exposure.

Author reply R3.3: This is a good point as we had not previously provided direct evidence of the increase in charge carrier concentration upon oxidation. We have now performed Hall effect measurements on 1% Cl doped $\text{CsSnI}_{3-x}\text{Cl}_x$ perovskite thin films. As we can see in Figure R2 (below), before doping, the initial charge carrier density was $2.38 \times 10^{18} \text{ cm}^{-3}$, and after air exposure, the charge density increased to $1.06 \times 10^{19} \text{ cm}^{-3}$. This indicates that our doping method based on the oxidation of Sn^{2+} to Sn^{4+} does achieve charge carrier densities suitable for thermoelectrics.

Our action R3.3: We have added Hall measurement details in the revised Supporting Information section 15 (not repeated here), and a summary of this in the methods section of the main text (not repeated here). We also replaced Figure 4e in the main text with Figure R2 (below).

We have added some discussion of the result on page 10 of the main text: “To verify this hypothesis, we used Hall measurements to determine the charge carrier concentration as a function of air exposure, showing an increase with air exposure from $2.38 \times 10^{18} \text{ cm}^{-3}$ to $1.06 \times 10^{19} \text{ cm}^{-3}$ after 12 minutes. Meanwhile, the Hall mobility decreases from an initial value of $76.1 \text{ cm}^2 \text{ V}^{-1} \text{ s}^{-1}$ to $50.1 \text{ cm}^2 \text{ V}^{-1} \text{ s}^{-1}$ after oxidation.”

Figure R2. Charge carrier density and Hall mobility of CsSnI_{3-x}Cl_x perovskite thin films (1% Cl-doped) plotted as a function of air exposure time.

Comment_R3.4: What is the mobility of the samples before and after the air exposure?

Author reply R3.4: Thanks for pointing out this. We have addressed this in response 3.3, where we obtained charge carrier densities and Hall mobilities as a function of air exposure through Hall measurements. In summary, the Hall mobility decreased from 76.1 cm² V⁻¹ s⁻¹ to 50.1 cm² V⁻¹ s⁻¹ with 12 minutes of air exposure. The trend of mobility decreasing is due to the defects introduced from doping processes.

Our action R3.4: Our actions are summarized in response R3.3

Comment_R3.5: When the author measures the electrical properties of sample under nitrogen atmosphere, the electrical conductivity also cannot stay at a constant as a function of time. The oxidation of Sn²⁺ to Sn⁴⁺ may be also caused by the interaction between Sn ions and halide ions or other aspects. Without comparison, it is not very convincing to conclude that air exposure is the only reason for the oxidation of Sn²⁺ to Sn⁴⁺.

Author reply R3.5: We appreciate the comments. Firstly, we agree that ion migration should always be considered in halide perovskite semiconductors. To explore the ion migration in CsSnI_{3-x}Cl_x, we fabricated perovskite solar cells with an architecture of ITO/CsSnI_{3-x}Cl_x/C₆₀/Bathocuproine (BCP)/Ag. The perovskite layer (270nm) was deposited by vacuum thermal evaporation (SLS method), and the 20 nm of C₆₀ acting as electron transport layer was also deposited by vacuum evaporation. 5nm of BCP was thermally evaporated as a buffer layer before depositing Ag electrodes (80nm). The unoxidised sample was encapsulated in a glovebox before being transferred to the test facility without any further air exposure. The oxidized sample was intentionally exposed to air prior to encapsulation. The J-V

characterization was performed under AM1.5G solar simulator illumination with a Keithley 2400 sourcemeter. As shown in Figure R4, the device without air oxidation exhibited hysteresis with different open circuit voltages (V_{oc}) of 0.48 V and 0.54 V for the forward and reverse sweeps respectively. The device with 10 minutes air exposure exhibited no significant hysteresis and a V_{oc} of 0.38 V. We can therefore confirm that the effects of ion migration are only observed in devices made from unoxidized $\text{CsSnI}_{3-x}\text{Cl}_x$ perovskite, and not in the high conductivity films which exhibit high ZT.

Figure R4. J-V characteristics of the $\text{CsSnI}_{3-x}\text{Cl}_x$ perovskite solar cells. Left: measurement of sample without prior air exposure (no oxidation). Right: measurement of a sample intentionally exposed to air for 10 minutes.

Secondly, we explored whether ion migration has any effect on the electrical conductivity. For comparison, we performed the electrical measurement in vacuum of the samples without air oxidation (Figure R5, below). As discussed above, there is an observable ion migration in solar cells of the unoxidised samples. Figure R5 shows three measurements on these samples over a period of 6 hours, during which the electrical conductivity remains constant. This shows that ion migration does not have an impact on the measurement of electrical conductivity in our samples.

Figure R5. Electrical conductivity of CsSnI_{3-x}Cl_x perovskites measured in vacuum.

Finally, we should mention that in our main text, when we measure the electrical conductivity of the samples stored in glovebox, the electrical conductivity increases with time. The reason is that there is tiny amount of water and oxygen in glovebox (<1 ppm) as well as residual amounts of solvent. This is not present for the measurements in vacuum discussed above.

Comment_R3.6: Ref 4 should be updated, since the highest PCE is above 24%.

Author reply R3.6: Thanks for pointing out this. At the time of writing, the highest PCE reported by NREL is 25.2%.

Our action R3.6: We have edited the main text on page 2: “Halide perovskite have been recognized as promising photovoltaic materials achieving a power conversion efficiency exceeding 25%.” We updated Ref 4: Best Research-Cell Efficiency Chart, <https://www.nrel.gov/pv/cell-efficiency.html> (2019)

Comment_R3.7: On page 4, the claim of doping Cl in the top 10 nm layer is difficult, since it will change with time by ion migration.

Author reply R3.7: Thanks for raising this issue. We note that one of the main reasons for ion migration in perovskite solar cells is the large electric fields generated between the electrodes ($\sim 0.5\text{V}$ over $\sim 100\text{nm}$ $\approx 5\text{ MV/m}$). In our case, mild electric fields ($\sim 500\text{ V/m}$) are applied in-plane, but the chlorine concentration gradient is out-of-plane. We therefore don't expect rapid ion migration caused by electric fields.

Nonetheless, to explore the possibility of ion migration with time, we repeated our XPS depth profile on a mixed halide perovskite films that was stored in a glovebox for an extended period of time before measurement. As shown in Figure R6 (below), this sample showed Cl in just the top 7.5 nm of the films, in line with the measurement on a fresh sample presented in the original manuscript. Therefore, ion migration does not seem to be altering the vertical composition of our films.

Figure R6. XPS depth file of 1% Cl doped $\text{CsSnI}_{3-x}\text{Cl}_x$ films kept in glovebox for 2 hours. Left: Cl 2p, middle: Cs 3d and right: I 3d.

Comment_R3.8: The exists of Cl layer will only slow down the oxidation as mentioned by the author, it can not solve the problem.

Author reply R3.8: The stability issue of perovskite materials has been investigated since 2012: including extrinsic factors (environmental) and intrinsic factors (material itself). Meng et al., [Nature Communications 9, 5265 (2018)] provide a good review of the stability challenges, noting that the environmental problems arising mostly from oxygen and water can be resolved by encapsulation of devices. Considering the material itself, structural instability and ion migration are identified by Meng et al. as the main factors. Several methods can improve the intrinsic stability, such as composition tuning (cation and/or alkali doping), passivating the grain boundaries, molecular additives and perovskite dimensional engineering (i.e. going from 3D to 2D layered materials). To the best of our knowledge, no paper to date has eliminated the instabilities, but much progress has been made. Correspondingly, our method cannot totally eliminate instabilities in tin based perovskites, but we have demonstrated a strategy for improving stability (i.e. Cl-doping). We feel that it is beyond the scope of this paper to explore all possible pathways for further stabilizing the material.

Comment_R3.9: On page 6, the reason for the different conductivity change behavior is unclear and should be explained or supposed.

Author reply R3.9: Our results show that film morphology plays a key role in stability of films and the potential for high conductivity. Different conductivity and degradation behaviours in air and nitrogen atmospheres were observed between films made by the three deposition methods. These three types of film have different dominant grain orientations and we note that grain orientation has previously been shown to have a large impact on degradation rates in air [Ma et al., *Nanoscale*, **2019**, 11, 170]. Specifically, co-evaporated films with dominant (220) orientation show a good air stability but low electrical conductivity. Sequentially deposited films with dominant (202) orientation show a poor air stability despite larger grain sizes than the other films, but actually show higher electrical conductivity. SLS films combining co-evaporation (for the seed layer) with sequential deposition have

a mixture of (220) and (202) grain orientations. These films show a comparable electrical conductivity to sequentially deposited films, smaller grain sizes and intermediate air stability.

Our action R3.9: We have added discussion of this point on page 6, and now it reads:” Co-evaporated films with dominant (220) orientation therefore show the best air stability but lowest electrical conductivity. Sequentially deposited films with dominant (202) orientation show a poor air stability despite their larger grain sizes, but do have higher electrical conductivity. Previous work has shown that grain orientation can have a significant effect on degradation rates of halide perovskite films [Ma et al., *Nanoscale*, 2019, 11, 170] and this is likely to be the case here.”

Comment_R3.10: Page 9, the Sn⁴⁺ on the surface will protect the bottom from oxidation is hard to understand. It is not the case of Al₂O₃ on Al. This is opposite to the observation in page 10.

Author reply R3.10

Thanks for this comment. We agree that we are not presenting the case of Al₂O₃ on Al where there is a permanent protection of the underlying metal by the oxide. Also, in our case, it is not Sn⁴⁺ at the surface which is slowing down the oxidation of underlying layers, it is the presence of the Cl-dopants that is doing this. Nonetheless it is the surface Sn⁴⁺ which provides the charge carriers for the unoxidised bulk, and these surface Sn⁴⁺ states are present in the Cl-doped films. Cl-doping of the top layer therefore does not prevent oxidation of the top layer, but slows down oxidation of the lower layers. We also note that even the Cl-doped layer is not a permanent barrier to oxidation of the underlying film, but it does significantly slow it down.

Our action R3.10:

To make our case more clearly and to avoid confusion, we have edited our main text on page 9, and it now reads: “From this, we can conclude that the top surface layer of the mixed CsSnI_{3-x}Cl_x acts as a sacrificial layer where initial oxidation occurs. This layer provides hole doping to the bulk (*vide*

infra) from the surface Sn⁴⁺ species. This mechanism, which separates the doping layer from the transport region, minimises the structural impact of doping on charge mobility, and enables our mixed halide perovskite structure to present high electrical conductivity whilst retaining a reasonable degree of air stability.”

Reviewers' comments:

Reviewer #1 (Remarks to the Author):

Authors have significantly improved the quality of the paper and have provided satisfactory answers to most of the comments.

While I agree with the authors that the work of Saini et al. is preliminary in nature but there are interesting aspects to notice. Saini et al. obtained ZT of 0.13 at room temperature in contrast to present work where ZT of 0.14 was obtained at 345 K. I think it is fair to mention the corresponding temperature where the ZT value is mentioned because it is a temperature dependent value. The stability of the present devices are extremely poor. Authors should provide some future guidelines for improving the stability of these unstable perovskites. And what are the advantages of developing such class of thermoelectrics over traditional materials.

Reviewer #2 (Remarks to the Author):

I suggest acceptance of the manuscript since the authors have made appropriate revisions.

Reviewer #3 (Remarks to the Author):

In general, the authors addressed the comments. However, this article should also include a figure or table to compare its new data with previous reported data about the application of similar perovskites materials in thermoelectrics. Additionally, for Cu based perovskites in the self-doping mechanism, the valence state of Cu is related to the atomic ratio of halide element (DOI: 10.1021/acs.inorgchem.5b01896). Does the Sn²⁺/Sn⁴⁺ ratio also partially depend on the ratio of Cl/I?

Reviewers' comments:

Reviewer #1 (Remarks to the Author):

General comment_R1: Authors have significantly improved the quality of the paper and have provided satisfactory answers to most of the comments.

Reply to general comment_R1: We thank referee for reviewing our revised manuscript.

Comment_R1.1: While I agree with the authors that the work of Saini et al. is preliminary in nature but there are interesting aspects to notice. Saini et al. obtained ZT of 0.13 at room temperature in contrast to present work where ZT of 0.14 was obtained at 345 K. I think it is fair to mention the corresponding temperature where the ZT value is mentioned because it is a temperature dependent value.

Reply to comment_R1.1: Thanks for this suggestion. We have added the temperature for the mentioned ZT value where it appears in the introduction of the revised manuscript.

Comment_R1.2: The stability of the present devices are extremely poor. Authors should provide some future guidelines for improving the stability of these unstable perovskites. And what are the advantages of developing such class of thermoelectrics over traditional materials.

Reply to comment_R1.2: Thanks for this suggestion. To improve the stability of the tin based perovskites, one possibility is to improve the stability of Sn^{2+} , by adding a strong reductant with favourable Goldschmidt tolerance.[e.g. Europium (Eu) as reported in Wang et al., Science 363, 265-270 (2019)] A second possibility is to use two dimensional (2D) perovskite layers to improve the structural stability, including layered and Ruddlesden-Popper structures.[Chem. Mater., 28, 8, 2852-2867 (2016)] Thirdly, one could use further mixed halide, mixed cation and mixed metal approaches that have been demonstrated in solar cells [e.g. Chen et al., Nat. Commun. 10, 16 (2019)]. Finally, we suggest using surface passivation layers or barrier layers to improve the air stability [Wang et al., Science 365, 687-691 (2019)] or full encapsulation [Matteocci et al., Nano Energy 30, 162-172].

One advantage of using halide perovskite materials are that they are elementally abundant compared to conventional low temperature thermoelectric materials such as Bi_2Te_3 . Secondly, these perovskite materials could be used as thin film thermoelectric generators for near room temperature applications with mechanical flexibility, solution processability and large area scalability. It is also feasible that

high ZT near room temperature range could be achieved within this family of materials due to low thermal conductivity, high Seebeck coefficients and high charge mobility in this temperature range.

Our action_R1.2: We have extended our conclusions with the guidelines for improving stability and potential advantages of these materials. The addition reads: “The potential advantages of halide perovskite materials for thermoelectrics are elemental abundance as well as mechanical flexibility, solution processability and large area scalability. Finally, we note that the stability of this tin based perovskite material could be further improved by adding a strong reductant with favourable Goldschmidt tolerance into the structure, pursuing layered structures, surface passivation or adopting mixed metal approaches.”

Reviewer #2 (Remarks to the Author):

I suggest acceptance of the manuscript since the authors have made appropriate revisions.

Reply: We would like to thank the referee for reviewing our revised manuscript.

Reviewer #3 (Remarks to the Author):

In general, the authors addressed the comments. However, this article should also include a figure or table to compare its new data with previous reported data about the application of similar perovskites materials in thermoelectrics. Additionally, for Cu based perovskites in the self-doping mechanism, the valence state of Cu is related to the atomic ratio of halide element (DOI: 10.1021/acs.inorgchem.5b01896). Does the $\text{Sn}^{2+}/\text{Sn}^{4+}$ ratio also partially depend on the ratio of Cl/I?

Reply: We thank the referee for reviewing our revised manuscript. We have added a summary table in the revised manuscript for comparing our data with the perovskite based thermoelectric materials.

For the self-doping mechanism discussed in $\text{MA}_2\text{CuCl}_{4-x}\text{Br}_x$ perovskites (DOI: 10.1021/acs/inorgchem.5b01896), the presence of Cu^+ is related to the Br/Cl ratio confirmed by X-ray photoelectron spectroscopy (XPS) of the Cu $2p$ peak. In our work, we investigated Sn $3d$ chemical states by XPS (supporting information 7, Figure S9f). As we can see from Figure S9, the $3d_{5/2}$ peak shift from 485.8 eV at 0% Cl to higher binding energy (486.3 eV) at 1% Cl, which is consistent with

the $\text{Sn}^{2+}/\text{Sn}^{4+}$ ratio increasing with the Cl/I ratio. This is in-line with the observations on Cu based perovskite. However, it is difficult to fully resolve differences in Sn-oxidation states by conventional XPS due to the small energetic shifts, which is why we investigated the oxidation state of tin by Auger electron spectroscopy as a function of depth. This enabled a more rigorous and unambiguous identification of oxidation state.

Our action: We have added a table of the performance summary of halide perovskite based thermoelectric materials in Supporting Information section 17. We have also add the reference (DOI: 10.1021/acs/inorgchem.5b01896) in the revised manuscript where we discuss the self-doping process of Sn^{2+} . (Reference 44)

Table S4. Summary of the thermoelectric properties in the halide perovskite materials

	Temperature (K)	materials/ methods	dopant	Figure of merit, ZT	reference
$\text{CH}_3\text{NH}_3\text{SnI}_3$	295	bulkcrystals	N/A	$\sim 10^{-3}$	15
$\text{CH}_3\text{NH}_3\text{PbI}_3$	295	bulkcrystals	Light	$\sim 10^{-7}$	
CsSnI_3	320	nanowires	N/A	0.11	16
CsSnI_3	295	thin films	N/A	0.137	17
$\text{CH}_3\text{NH}_3\text{SnI}_3$	300	simulation	n-type	1.7	18
			p-type	1.1	
$\text{CH}_3\text{NH}_3\text{PbI}_3$	300	simulation	n-type	1.2	19
			p-type	0.8	
$\text{C}_6\text{H}_4\text{NH}_2\text{CuBr}_2\text{I}$	363	simulation (κ) thin films	N/A	0.22	19
CsSnI_3	345	thin films	SnCl_2	0.14	Our work

REVIEWERS' COMMENTS:

Reviewer #1 (Remarks to the Author):

The authors have responded to all the amendments. The manuscript is improved and can be published as it is.

Reviewer #3 (Remarks to the Author):

The authors has addressed most of my concerns. Now I have no further comments to the manuscript.

Reviewer #1 (Remarks to the Author):

The authors have responded to all the amendments. The manuscript is improved and can be published as it is.

Response to Reviewer #1: We thank you for your contribution to this review process.

Reviewer #3 (Remarks to the Author):

The authors has addressed most of my concerns. Now I have no further comments to the manuscript.

Response to Reviewer #3: We thank you for your contribution to this review process.